# Provably-secure quantum randomness expansion with uncharacterised homodyne detection

Chao Wang [1,4], Ignatius William Primaatmaja[1,2,4], Hong Jie Ng [1], Jing Yan Haw[1], Raymond Ho[1], Jianran Zhang[1], Gong Zhang[1] & Charles Lim [1,2,3] ✉

Quantum random number generators (QRNGs) are able to generate numbers that are certifiably random, even to an agent who holds some side information. Such systems typically require that the elements being used are precisely calibrated and validly certified for a credible security analysis. However, this can be experimentally challenging and result in potential side-channels which could compromise the security of the QRNG. In this work, we propose, design and experimentally demonstrate a QRNG protocol that completely removes the calibration requirement for the measurement device. Moreover, our protocol is secure against quantum side information. We also take into account the finite-size effects and remove the independent and identically distributed requirement for the measurement side. More importantly, our QRNG scheme features a simple implementation which uses only standard optical components and are readily implementable on integrated-photonic platforms. To validate the feasibility and practicability of the protocol, we set up a fibre-optical experimental system with a home-made homodyne detector with an effective efficiency of 91.7% at 1550 nm. The system works at a rate of 2.5 MHz, and obtains a net randomness expansion rate of 4.98 kbits/s at $10^{10}$ rounds. Our results pave the way for an integrated QRNG with self-testing feature and provable security.

Random number generators (RNGs) are the basic building block of many computing methods and digital solutions in use today, e.g., in simulation, optimisation, cryptography, and gambling. Ideally, the output of an RNG should be *uniform* and *unpredictable*. The first property requires all outputs are equally likely and the second stipulates that no observer can do better than a random guess even with side information about the device. Indeed, the latter property is especially important when dealing with digital technologies like secure communications, block-chain, and digital lottery, where privacy and information security are critical.

Quantum processes are excellent sources of randomness due to their intrinsic probabilistic nature. In particular, by tapping on the uncertainty of quantum measurements, one can, in principle, devise quantum random number generators (QRNGs) which are perfectly uniform and unpredictable[1–3]. The standard approach uses a model-based approach, where the underlying probability model is based on the certain trusted quantum measurement process. However, while this approach presents a straightforward way to quantify the amount of extractable randomness, it is prone to implementation deviations (for a concrete example, we refer the reader to ref. [4]). In particular, the

[1]Department of Electrical & Computer Engineering, National University of Singapore, Singapore, Singapore. [2]Centre for Quantum Technologies, National University of Singapore, Singapore, Singapore. [3]Global Technology Applied Research, JPMorgan Chase & Co, Singapore, Singapore. [4]These authors contributed equally: Chao Wang, Ignatius William Primaatmaja. ✉e-mail: charleslim.research@gmail.com

**Table 1 | Features of our proposed QRNG protocol as compared to the features of existing protocols**

| References | On-chip compatibility | Categories | Uncharacterised source | Uncharacterised measurement | Finite-size analysis | Remove i.i.d. assumption | Side-Information considered |
|---|---|---|---|---|---|---|---|
| Refs. [56,57] | ✗ | TD | ✗ | ✗ | ✗ | ✗ | none |
| Refs. [18,20,58,59] | ✓ | TD | ✗ | ✗ | ✗ | ✗ | classical |
| Refs. [19,60] | ✓ | TD | ✗ | ✗ | ✓ | ✓[a] | quantum |
| Refs. [61-63] | ✓ | Semi-DI | ✓[b] | ✗ | ✓ | ✗ | quantum |
| Ref. [64] | ✗ | Semi-DI | ✓ | ✗ | ✓ | ✓ | quantum |
| Refs. [65,66] | ✓ | Semi-DI | ✓ | ✗ | ✓ | ✓ | quantum |
| Ref. [67] | ✗ | Semi-DI | ✓[b] | ✓ | ✓ | ✓ | classical |
| Ref. [68] | ✓ | Semi-DI | ✓[b] | ✓ | ✓ | ✓ | classical |
| Ref. [69] | ✓ | Semi-DI | ✓[b] | ✓ | ✗ | ✗ | classical |
| Refs. [70,71] | ✗ | Semi-DI | ✓[c] | ✓ | ✗ | ✗ | classical |
| Ref. [72] | ✓ | Semi-DI | ✓[b] | ✓ | ✓ | ✓[d] | classical |
| Ref. [73] | ✗ | Semi-DI | ✓[e] | ✓[e] | ✓ | ✗ | classical |
| Refs. [74-76] | ✗ | Semi-DI | ✗ | ✓ | ✗ | ✗ | classical |
| Ref. [5] | ✗ | DI | ✓ | ✓ | ✓ | ✓ | quantum[f] |
| Ref. [8] | ✗ | DI | ✓ | ✓ | ✓ | ✓ | classical |
| Refs. [6,7,9] | ✗ | DI | ✓ | ✓ | ✓ | ✓ | quantum |
| This work | ✓ | Semi-DI | ✗ | ✓ | ✓ | ✓ | quantum |

In this Table, protocols that are compatible with on-chip implementation refer to protocols which are built on photodiodes, e.g., homodyne detection, which had been implemented on the Photonic Integrated Circuits (PICs) (for example, see refs. [11-14]). On-chip single-photon detection has also been demonstrated recently[80,81] but it has not been widely adopted yet. Moreover, cooling is typically required in single-photon detection to achieve a desirable dark count level, which leads to additional system complexity and space usage.

*Semi-DI* Semi-Device-Independent scheme, *DI* Device-Independent scheme.

[a] By assuming Gaussianity and stationarity of the noisy processes.
[b] With additional assumption on the input energy.
[c] With assumption on the overlap of the states.
[d] With additional assumptions of i.i.d. source and channel.
[e] With additional assumption on the system dimension.
[f] The randomness certification presented in this paper contained some errors as clarified in subsequent papers[77,78]. The experiment was then re-analysed in ref. [79] where it was shown that randomness against quantum side information was indeed attained.

model may not capture the actual physical process due to unexpected device changes and the amount of extractable randomness can be overestimated. Crucially, this could lead to catastrophic outcomes when the device is used for cryptography, for example.

An elegant solution is to consider new forms of QRNGs which provide certifiable randomness based on minimal assumptions about the underlying quantum measurement process. This is made possible by exploiting the unique correlations established by quantum measurements on entangled systems. The best example is the Device-Independent (DI) QRNG[1,5–9]. However, in view of the demanding experimental requirements for a loophole-free observation of non-local correlations[5,7,10], it is generally believed that the first practical application of such DI QRNGs will likely be Randomness Beacons[7,8].

There are other QRNGs that make reasonable assumptions about the system and require only a partial characterisation of the device. These schemes provide a system performance improvement in terms of the implementation complexity and the random number generation rate when compared to DI QRNGs. Due to the partial characterisation feature, this class of QRNGs is often called semi-DI QRNG. A comparison of different QRNG studies is listed in Table 1.

For practical semi-DI randomness generations, the following features are highly desirable in practical applications. Firstly, the security of the randomness generation should rely on only a few justifiable assumptions on the system operation and its critical components. This would ensure that these QRNGs remain secure even in the presence of unexpected device changes. Secondly, it should also provide a relatively high randomness generation rate. Finally, the QRNG should be cost-effective and have small footprints. The latter would be essential in a wide range of applications: from handheld devices to Internet-of-Things.

In this regard, balanced homodyne detection offers distinct advantages in practical randomness generation. Firstly, as balanced homodyne detectors simply consist of a pair of photodiodes and some electrical components, they are readily implementable on integrated-photonic platforms[11–14]. Hence, QRNGs that are based on homodyne detectors have a unique practical advantage in terms of the cost-effectiveness, compactness and system stability. Secondly, homodyne detection works at room temperature and no additional cooling is needed. This, again, reduces the system complexity and eliminates the extra requirement for space consumption.

Unfortunately, due to many practical limitations, real homodyne detectors often deviate from an ideal quadrature measurement – which is the standard theoretical model for a balanced homodyne detection. Firstly, modelling homodyne detectors as perfect quadrature measurements of the input optical field requires the local oscillator (LO) to operate at the high intensity limit[15,16], which may not be the case in the actual implementation. In addition, implementing the perfect quadrature measurement would also require perfect photon number subtraction which is non-trivial in the presence finite common-mode rejection ratio (CMRR) and imbalance drifts. Moreover, in contrast to perfect quadrature measurements, practical homodyne detectors are subjected to electronic noise, LO intensity fluctuations, finite detection range, etc. While there are theoretical studies on how to account for these imperfections in the model (for example, the standard theoretical treatment for electronic noise is to model it as an independent noise[17,18] with Gaussian and stationary nature[18,19]), the model demands an accurate characterisation of each imperfection. Not only is this task technically demanding, there is actually a danger of false precision with the model-based approach as the quality of the homodyne detector may degrade over time. In that

case, this would invalidate the model and hence, the randomness generated by the homodyne detection may be overestimated.

Additionally, it has been pointed out that finite bandwidths of practical homodyne detector may result in correlations among successive rounds of the QRNG operation[19,20]. In this case, the experimental rounds in practice would be unlikely to exhibit a completely independent and identically distributed (i.i.d.) behaviour. In particular, comparing to the quantum state generation part, the homodyne detector with a shot-noise-limited performance generally has a more restricted working bandwidth[17,21]. Therefore, it is of great interest to devise a randomness certification that can mitigate (if not fully remove) the i.i.d. assumption for the system operation. Moreover, as any QRNG protocols have to be executed in a finite number of rounds, finite-size effects (such as statistical fluctuations) should be taken into consideration. This is especially important for semi-DI and DI QRNGs, whose randomness certification relies on experimental statistics, which necessarily entail statistical fluctuations.

In this work, we propose a novel semi-DI QRNG protocol based on homodyne detection and certify its security against quantum side information. Given the challenges with modelling the homodyne detector, our proposed framework treats it as a black-box quantum measurement which sidesteps the demanding characterisation requirement of the model-based approach. Importantly, our framework does not require any i.i.d. assumption on the measurement device which protects the security of the protocol against potential correlations shared between different rounds. Moreover, the proposed protocol is composably secure[22–24], which guarantees that the random numbers produced by our protocol can be securely used for cryptographic applications. Furthermore, we show that our protocol can produce more randomness than that consumed (for choosing the settings for the devices) in the protocol. As such, our protocol is a *quantum randomness expansion* (QRE) protocol.

## Results

### Protocol description

We shall now present our proposed randomness generation protocol. The protocol that we consider is a prepare-and-measure (P&M) protocol with an uncharacterised measurement device. To illustrate the protocol, it is convenient to consider a device that consists of two parts: a trusted source of quantum states (which we assume to be operated by Alice) and an uncharacterised measurement device (which is operated by Bob). As such, the protocol that we consider is a self-testing protocol in which the working of Bob's measurement

device is not assumed a priori, but could be verified during the protocol using the spot-checking scheme in which every round is randomly assigned to be a generation or test round.

To that end, suppose that during the test round, the device plays a P&M game $\mathcal{G}$. A P&M game can be thought of as a P&M analogue of the more well-known non-local games in the context of Bell nonlocality[25,26] and device-independent protocols. In a P&M game, Alice receives a random input $x$ from a pre-defined set $\mathcal{X}$ and then prepares the state $|\psi_x\rangle$ from the set of states $\mathcal{S}_{\mathcal{X}}$. Similarly, Bob receives an input $y$ from a pre-defined set $\mathcal{Y}$ and uses it as his measurement setting. Let us suppose that Alice and Bob receive those inputs with probability $q(x, y)$ which is fixed for a given game. For each pair of inputs $x$ and $y$, the game $\mathcal{G}$ defines the winning outcome $b_{xy} \in \{0, 1\}$. For a given round, the device wins the game when Bob outputs the winning outcome; otherwise, the device loses the game. In the Methods section, we present a systematic method to choose the winning outcome $b_{xy}$ as well as the probability of choosing each pair of inputs $q(x, y)$.

The protocol that we propose is given in Box 1

In the experiment reported in this work, we consider $\mathcal{X} = \{0, 1, 2, 3\}$, the set of states $\mathcal{S}_{\mathcal{X}} = \{|\alpha e^{ix\pi/2}\rangle : x \in \mathcal{X}\}$ and $\mathcal{Y} = \{0, 1\}$. Furthermore, the honest implementation corresponds to homodyne detection with its LO's phase set to $\varphi = \pi/2$ when $y = 0$ and $\varphi = 0$ when $y = 1$. The measurement device would then output $b = 0$ if the result of the homodyne measurement is positive; else, it would output $b = 1$. The quantum states in this case correspond to quadrature phase shift keying (QPSK) modulation format, which is compatible with standard optical modulation techniques. Remarkably, it is straightforward to generalise the protocol to include more states or more measurement settings, e.g. quadrature amplitude modulation (QAM).

### Security framework

We shall now consider the security of our protocol. Here, we consider a framework in which the measurement device is uncharacterised and hence, when analysing the security of the proposed protocol, we shall treat Bob's measurement as a set of abstract measurement operators. In particular, we do not assume that Bob's measurement device behaves independently and identically for each round.

Likewise, we do not assume that the quantum channel faithfully transmits the quantum states sent by Alice, nor it behaves independently and identically for each round. As we do not limit the dimension of the Hilbert space of the channel output, we can model any quantum channel by an isometry $U$ which preserves the inner-product of the states prepared by Alice's trusted source.

---

## BOX 1

# Our proposed QRNG protocol

*Arguments:*

$n$—the number of rounds

$\gamma$—testing probability

$\mathcal{X}$—the set of possible inputs for Alice

$\mathcal{S}_{\mathcal{X}}$—the set of states that Alice can prepare

$\mathcal{Y}$—the set of possible inputs for Bob

$\mathcal{G} = \{(b_{xy}, q(x, y)) : x \in \mathcal{X}, y \in \mathcal{Y}\}$—the P&M game

$\omega$—the expected probability of winning the game $\mathcal{G}$

$\delta$—the width of the confidence interval for the winning probability

Ext—a strong quantum-proof seeded extractor

**S**—random seed for randomness extraction

*Protocol:*

1. For each round $i \in [n]$: do Step 2 to 4
2. Set $C_i = \perp$ and randomly choose $T_i \in \{0, 1\}$ such that $\Pr[T_i = 1] = \gamma$.
3. If $T_i = 0$, set $X_i = 0$ and $Y_i = 0$. Else, set $X_i = x \in \mathcal{X}$ and $Y_i = y \in \{0, 1\}$ with probability $q(x, y)$.
4. Alice prepares the coherent state $|\psi_{X_i}\rangle \in \mathcal{S}_{\mathcal{X}}$ depending on her input $X_i$. Bob sets his measurement setting to $Y_i$ and records the output $B_i \in \{0, 1\}$. If $T_i = 1$, they would set $C_i = 0$ if $B_i \neq b_{X_i Y_i}$ and $C_i = 1$ if $B_i = b_{X_i Y_i}$.
5. If $|\{i : C_i = 0\}| > n\gamma(1 - \omega + \delta)$, then abort the protocol. Otherwise, we accept the protocol execution and preserve the data for further processing.
6. Apply a quantum-proof strong seeded extractor Ext using a uniformly chosen random seed **S**. Denote the output $\mathbf{Z} = \text{Ext}(\mathbf{B}, \mathbf{S})$. Since a strong extractor is used, the protocol outputs the concatenation $\mathbf{K} = (\mathbf{Z}, \mathbf{S})$.

Additionally, we also allow the adversary (or any agent trying to guess the output of the protocol), Eve, to have some pre-shared entanglement with Bob's uncharacterised device, but due to some technicality regarding the method used to certify the generated randomness, we assume that Eve does not obtain additional quantum side information when the protocol is executed. This assumption can be well justified for the setting considered in QRNG protocols where Alice and Bob are both inside the same secure location.

Finally, we also assume that the device is equipped with trusted and private random seed that is used to choose the inputs for each round as well as to perform seeded extraction. For a detailed discussion on the assumptions we make in the randomness certification, we refer the readers to the Methods section 'Randomness certification'.

The security of our protocol relies on the quantum-proof strong seeded extractor which guarantees that whenever the protocol is not aborted, the output string is close to an ideal random bit-string that is uniformly random and independent from any pre-shared quantum information held by the adversary as well as the initial random seed. Hence, we have to certify that our protocol produces enough randomness (measured in terms of the conditional smooth min-entropy of the raw string) before applying the randomness extraction. To that end, we adopt the framework of entropy accumulation theorem (EAT)[27–30]. Informally, the EAT states that when our protocol is not aborted, the conditional smooth min-entropy of the raw string given Eve's side information and the random inputs is at least

$$nh(\omega, \delta) - \mathcal{O}(\sqrt{n}). \tag{1}$$

Importantly, the leading term which scales linearly with the number of rounds $n$ can be evaluated by analysing a single-round of the protocol. The constant of proportionality $h(\omega, \delta)$ as well as the correction term $\mathcal{O}(\sqrt{n})$ can then be computed using a semi-definite-programming (SDP) technique introduced in ref. [31]. More precisely, we have the following theorem.

**Theorem 1.** (Entropy accumulation theorem (as modified from Lemma III.3 of ref. [30])) Let $\Omega$ denote the event in which our QRNG protocol is not aborted and $\rho^\Omega$ be the final state conditioned on this. Let $f(1-\nu)$ be an affine lower bound on the single-round conditional von Neumann entropy for any strategy that wins the game $\mathcal{G}$ with probability $\nu$. For fixed parameters $\epsilon_s, \epsilon_{EA}, \beta \in (0, 1)$, then either our QRNG protocol aborts with probability greater than $1-\epsilon_{EA}$ or

$$H_{\min}^{\epsilon_s}(\mathbf{B}|\mathbf{T}, \mathbf{X}, \mathbf{Y}, E)_{\rho^\Omega} > nf(1 - \omega + \delta) - \frac{1}{\beta}\left[1 - 2\log_2(\epsilon_{EA}\epsilon_s)\right] \\ - n\left[\beta V(\gamma, f) + \beta^2 K(\beta, \gamma, f)\right]. \tag{2}$$

The explicit expressions for the functions $f, V, K$ can be found in the "Methods" section 'Randomness certification'.

As Theorem 1 holds for any choice of $\beta$, as we can see in the Methods section, we can choose $\beta \propto 1/\sqrt{n}$ such that the correction term would scale with $\sqrt{n}$ as claimed earlier. We refer to the parameter $\epsilon_{EA}$ as the entropy accumulation error. As can be seen from Theorem 1, it quantifies our tolerance of encountering an event in which the protocol is not aborted but the lower bound (2) on the accumulated entropy does not hold.

Finally, with the lower bound on the conditional smooth min-entropy being established, we can use the quantum leftover hash lemma[32,33] to find the extractable length of the output string $\mathbf{Z}$, denoted by $\ell$. As the extractor seed $\mathbf{S}$ is part of the protocol output $\mathbf{K}$, the expected net randomness expansion rate $r_{net}$ is then defined as

$$r_{net} := \frac{\ell - \ell_{in}}{n}, \tag{3}$$

where $\ell_{in}$ is the expected amount of randomness used during the protocol to choose the settings of the device.

The details of randomness certification, the estimation of the input randomness can be found in "Methods" section 'Randomness certification' and 'Input randomness', respectively.

## Experimental implementation

In order to verify the feasibility of the proposed protocol, we set up a fibre-optic experimental system. The schematic diagram is shown in Fig. 1a.

Our experimental system is composed of two main parts, quantum state generation and quantum state measurement. In the quantum state generation, a laser diode emits continuous-wave (c.w.) laser with a central wavelength of 1550 nm and a linewidth of 50 kHz, which is split into two paths, one for quantum state preparation and the other as Local Oscillator (LO) for homodyne detection. In the signal path, an Intensity Modulator (IM) first curves the c.w. laser into pulses with pulse width of 4 ns each, for defining the temporal mode of the quantum states. Besides, the IM could also perform the intensity modulation for QAM-16 state generation. A Phase Modulator (PM) modulates the phase of the quantum states. Thereafter, the optical signals are attenuated to single-photon energy level with an optical attenuator, to finally generate the QPSK quantum states $\{|\alpha e^{ix\pi/2}\rangle\}$ where $x \in \{0, 1, 2, 3\}$, whose constellation diagram is shown in Fig. 1b.

In the quantum state measurement, a homodyne detector is deployed. To maximise the generated randomness, we developed a high-efficiency and low-noise fibre-coupled homodyne detector. To minimise the optical loss, we first adopt a pair of high-efficiency photodiodes (PDs) for photon detection. Moreover, we apply anti-reflection coated graded-index (GRIN) lens for the light coupling from optical fibre to the PDs. The overall efficiency of the PD including the coupling loss is measured to be 98.3% and 98.8%, respectively. The signal and LO are interfered in a balanced polarisation maintaining fibre-optical beam splitter (BS) before detection, providing good mode matching for a stable and efficient interference. After a careful balancing of the two arms, the photocurrents are subtracted and then amplified by a low-noise amplifier. The characterisation results of our homodyne detector are shown in Fig. 1 (d, e). The 3 dB bandwidth of our homodyne detector is ~72 MHz, and the clearance (shot noise to electronic noise ratio) is measured to be 16.94 dB with a 10 mW LO. Taking all the factors into consideration, the total effective efficiency of our homodyne detector is characterised to be 91.7%. The details of homodyne characterisation and modelling are provided in the "Methods" section 'Homodyne detector modelling and characterisation'.

In the actual experiment, the settings for quantum state preparation and measurement need to be optimised for a high net randomness expansion rate. For example, the randomness generation round could be chosen for most of the time (with a small testing probability $\gamma$) to obtain the optimal generation rate. This raises an issue with our AC-coupled systems such as the homodyne detector and the amplifiers for driving the modulators, where a DC-balanced data streams are preferred to eliminate potential signal distortions[34]. To mitigate this, we apply a complementary modulation scheme in our experiment where the protocol based on QPSK modulation is performed, as shown in Fig. 2a. With our scheme, the quantum state preparation is performed on two-mode coherent states, which are based on the two successive temporal modes. In this case, the modulation patterns for the state preparation and LO phase setting are naturally DC-balanced with any settings $x$ and $y$. Besides, the two temporal modes are modulated with a $\pi$ phase difference, while the LO phase settings are kept the same. As such, the expectation values of the individual quadrature measurements of the two temporal modes possess opposite values. Hence, the outputs of the homodyne detector are also naturally DC-balanced for all experimental settings. The quadrature measurement of the two-mode coherent state in this case

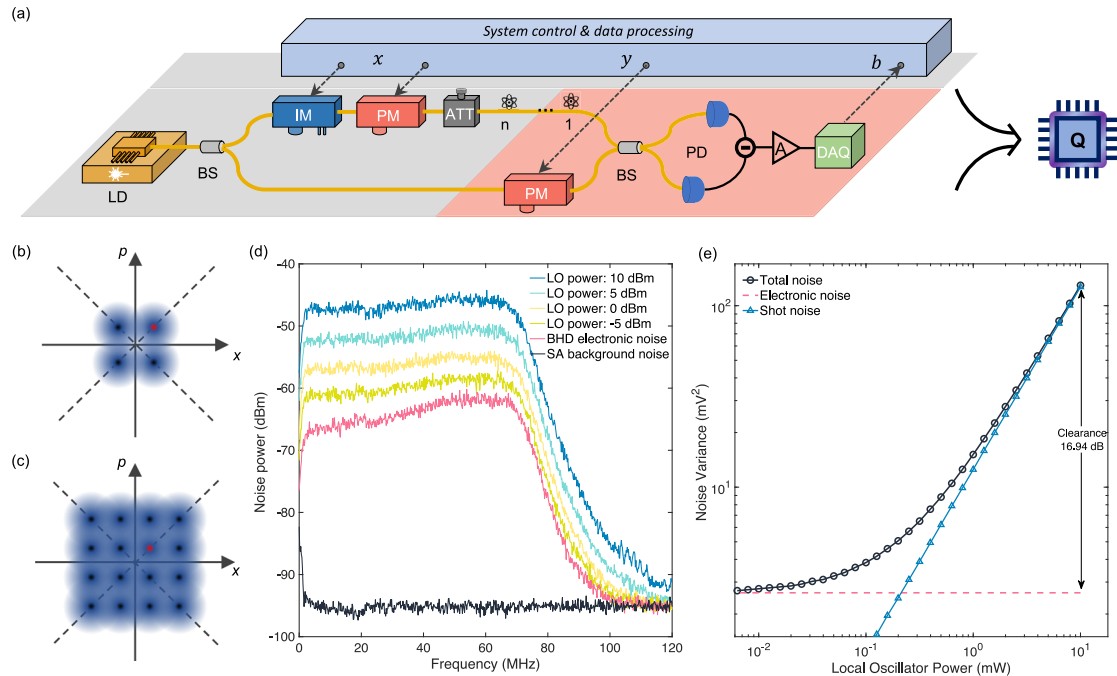

**Fig. 1 | Experiment Overview. a** Schematic of the experimental setup. The Laser Diode (LD) emits a continuous-wave laser, which is split into two parts by a Beam Splitter(BS). One is for quantum state preparation, and the other is for Local Oscillator (LO) for homodyne detection. In the signal path, an Intensity Modulator (IM) is used for pulse curving and intensity modulation, and a Phase Modulator (PM) is used for phase modulation. The optical signal is then attenuated to single-photon energy level by an attenuator (ATT). The final quantum states after modulation are in QPSK or QAM-16 format. In the LO path, a PM is deployed for basis choosing for the homodyne detection. The signal states and the LO are mixed on a balanced BS, and the photocurrent of two photodiodes (PD) are subtracted and further amplified. Finally, a data acquisition (DAQ) device samples the signal and obtain the data for analysis. **b, c** Constellation diagram for QPSK modulation and QAM-16 modulation, respectively. The blue circles represent the quantum states to be prepared by the transmitter. The circle with red centre represents the state used when the randomness generation round is chosen, and all the states are used for the testing rounds. The black dashed lines represent the two measurement bases in our protocol. For the convenience of illustration, we shift the phase of the states and measurements by $\pi/4$ comparing to the descriptions in the main text. This will not affect either the security analysis or the experimental results. **d, e** Homodyne detector characterisation. **d** Power spectrum of the homodyne detector from DC to 120 MHz. The 3 dB bandwidth is ~72 MHz. **e** Noise variance for different LO powers. A clearance of 16.94 dB is obtained with 10 mW LO input.

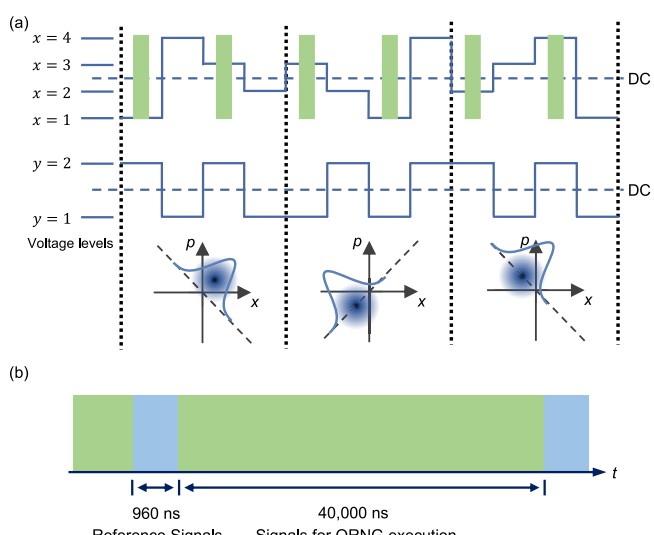

**Fig. 2 | Modulation Scheme. a** Illustration of the complementary modulation scheme. The blue curves represent the voltage levels applied for the signal and LO phase modulation. The green pulses represent the temporal modes for defining the two-mode coherent states. Regions divided by the black dashed lines represent the quantum state preparation and measurement for a single round of QRNG execution. The constellation diagrams illustrate the distributions of the quadrature measurement with given input settings. **b** Illustration of the time frame configuration. Reference signals for phase calibration are prepared and measured amongst the signals for QRNG execution.

is $q_t = \frac{1}{\sqrt{2}}(q_e - q_l)$, where $q_e$ and $q_l$ are the quadrature measurement value of the two individual temporal modes. For more details about the two-mode coherent states, please refer to the "Methods" section 'Two-mode coherent state'.

To faithfully implement the QRNG protocol, a fixed phase reference between the signal and LO is required. To this end, a feedback control for phase locking (not shown in Fig. 1a) is deployed, by analysing the statistics of the reference signals as well as the signals for QRNG execution. The illustration of the time frame configuration of our QRNG is shown in Fig. 2b.

At this point, we emphasise that since our QRNG protocol does not require any characterisation of the measurement device (i.e., the homodyne detector), our efforts in loss and noise reduction, phase locking, etc. do not affect the *soundness* of the randomness certification, but will certainly improve the performance in terms of randomness generation rate, the system stability (which is related to the *completeness* of the protocol), etc.

Based on Theorem 1, the quantum leftover hash lemma, and the definition of the expected net randomness expansion rate given in Eq. (3), we first simulate the performance of our proposed protocol. The simulated net randomness expansion rate with QPSK modulation and QAM-16 modulation are shown in Fig. 3a and b, respectively.

The parameters used in the experiment are listed in Table. 2, where the mean photon number $|\alpha|^2$ of quantum states in QPSK format, the probability of choosing test rounds $\gamma$ are optimised based on the system efficiency of our setup and the chosen security parameters, as shown in Fig. 4a, b. The relation of the expected net randomness expansion rate and the randomness consumed is shown in Fig. 4c.

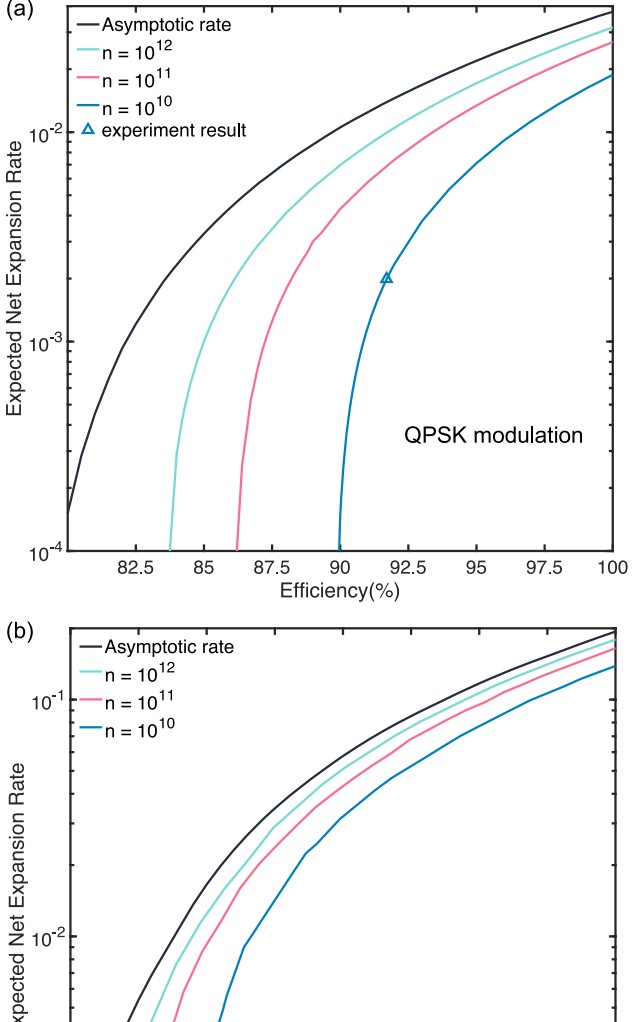

**Fig. 3 | Performance Analysis 1.** Expected net randomness expansion rate $r_{net}$ against system efficiency $\eta_{eff}$ for different number of rounds $n$ for (**a**) QPSK modulation and (**b**) QAM-16 modulation. Security parameters used for the simulation: $\varepsilon_{com}=1\times10^{-3}$, $\varepsilon_{sou}=1\times10^{-6}$ and $\epsilon_{EA}=1\times10^{-6}$.

According to the construction described in the "Methods" section 'Formulating a P&M game'., we formulate the P&M game $\mathcal{G}$ used in our QRNG protocol as shown in Table. 3, which determines the probability of choosing settings for Alice's input $x$ and Bob's input $y$ for test rounds and the scoring rules. Based on our model of the honest implementation (refer to the "Methods" section 'Two-mode coherent state' for details), we set $\omega = 0.59422$ and $\delta = 0.00189$ for the experiment,

which represents the expected winning probability, and the confidence interval for the winning probability, respectively.

We collect $n = 1\times10^{10}$ rounds of data in the experiment, and obtain an observed winning probability of $\omega_{obs} = 0.59443$, which is very close to the expected value. The number of rounds in which the players lost the game is 942820, which is within the acceptance range $n_{lost} \leq 946026$. Hence, the protocol execution is accepted, and we could certify a gross randomness generation rate (the randomness generated by the protocol per round) of at least 0.00455 bits per round can be obtained by running the protocol. Considering the randomness invested for determining whether a given round is a test round and the inputs of Alice and Bob ($x$ and $y$), the expected randomness consumption rate in our case is 0.00256 bits per round. Therefore, the expected net randomness expansion rate of our system is 0.00199 bits per round. The observed experimental results for test rounds are shown in Table 4.

Finally, we implement randomness extraction using Toeplitz hashing with the help of a random seed. Toeplitz hashing is a family of two-universal hash functions, and it has been shown to be a strong randomness extractor[32,35,36]. This means Toeplitz hashing not only extracts randomness from a weak entropy source, but also guarantees that the output string is independent of the seed. Thus, in this work, the seed required for randomness extraction is not considered as consumed randomness as the seed can be concatenated to the output due to the properties of a strong extractor.

We utilise a Zynq Ultrascale+ FPGA (XCZU28DR) to implement randomness extraction. We construct a Toeplitz matrix with the size of $45 \times 10,000$ to extract the random numbers from the raw data. To achieve a faster extraction speed, we further split the Toeplitz matrix into sub-blocks of size $45 \times 1000$ during the extraction. In total, the size of the raw data was 10.622 Gbits (we conservatively collected a bit more data than $1\times10^{10}$), and we extracted 47.8 Mbits of random numbers from it. The detailed implementation of Toeplitz hashing on FPGA is provided in Methods section 'Randomness extraction'.

## Discussion

In this section, we discuss the strengths and limitations of our work. Besides its high level of security due to its immunity to detection side channels, one of the main strengths of our protocol is its potential for miniaturisation. In the following, we discuss the feasibility of the proposed protocol to be implemented on silicon photonic integrated circuit (PIC), which is a leading platform for integrated-photonic applications with substantial advantages regarding miniaturisation, compatibility with CMOS microelectronics, and high-speed signal processing. The process design kits (PDKs) from major foundries can provide the key components on a single chip, with a decent performance stability for volume production[37–39].

By leveraging mature silicon-on-insulator (SOI) technology, the optical waveguide on silicon PIC is able to provide a low propagation loss and large integration density. The $2 \times 2$ beam splitter can be achieved by using either evanescent couplers or multimode interference (MMI) couplers.

For the high-speed optical modulation, the carrier depletion type modulators are available for quantum state preparation. While the high-speed Germanium photodetectors could be utilised for the homodyne detection on the optical quantum states.

## Table 2 | Parameters used in the experiment

| $n$ | $\varepsilon_{com}$ | $\varepsilon_{sou}$ | $\epsilon_{EA}$ | $|\alpha|^2$ | $\gamma$ | $\omega$ | $\delta$ |
|---|---|---|---|---|---|---|---|
| $1\times10^{10}$ | $1\times10^{-3}$ | $1\times10^{-6}$ | $1\times10^{-6}$ | $1.638\times10^{-2}$ | $1.587\times10^{-4}$ | 0.40578 | 0.00189 |

$n$: number of rounds. $\varepsilon_{com}$: Completeness error. $\varepsilon_{sou}$: Soundness error. $\epsilon_{EA}$: Entropy accumulation error. $|\alpha|^2$: The mean photon number of the quantum state. $\gamma$: The probability of choosing test round. $\omega$: The expected probability of winning the game. $\delta$: The width of the confidence interval for the winning probability.

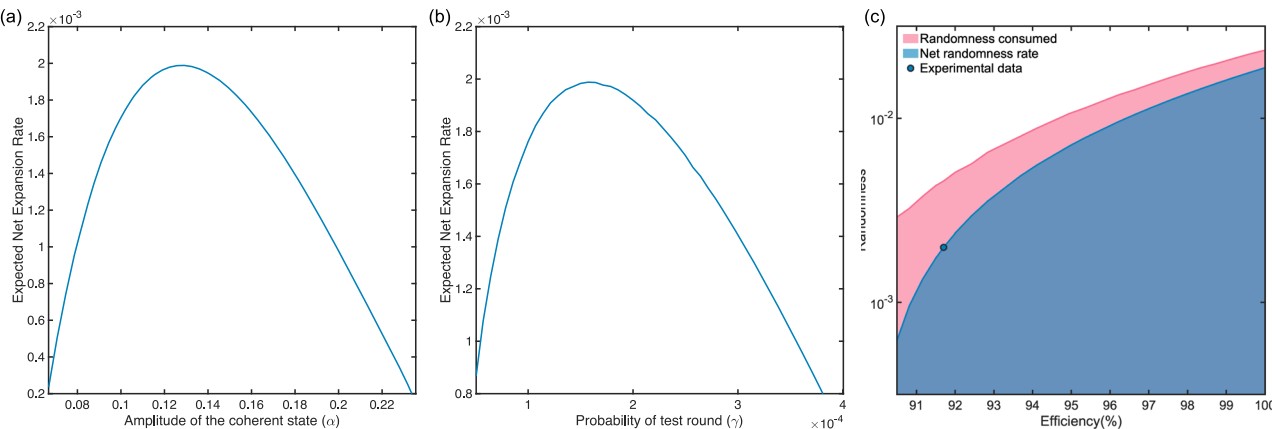

**Fig. 4 | Performance Analysis 2. a, b** The expected net randomness expansion rate of the QRNG versus the (**b**) amplitude of the quantum state $\alpha$ and (**c**) the probability of test rounds $\gamma$, with $\eta_{\text{eff}} = 91.7\%$. **c** The expected net randomness expansion rate and the randomness consumed versus the homodyne detection efficiency. The results are obtained by optimising the net expansion rate over $|\alpha|^2$ and $\gamma$. All plots are based on system parameters $n = 1 \times 10^{10}$, $\varepsilon_{\text{com}} = 1 \times 10^{-3}$, $\varepsilon_{\text{sou}} = 1 \times 10^{-6}$ and $\epsilon_{\text{EA}} = 1 \times 10^{-6}$.

The integrated laser, a critical component to our QRNG system, turns out to be a hurdle that impedes full system integration on a single silicon chip. This is because pure crystalline silicon lacks a direct bandgap precluding the possibility of monolithic silicon laser[38]. Fortunately, recent integrated laser technology based on packaging or heterogeneous integration technologies could be adopted to address this problem[37].

In this, one of our main objectives is to validate experimentally the key concept of the proposed protocol and this ultimately boils down to achieving sufficient effective efficiency of the measurement device to demonstrate positive net randomness expansion rate. To that end, we require high-efficiency PDs on silicon PICs. Fortunately, such highly efficient PDs are within the reach of current technology. For example, Globalfoundries offers PDs with >1 A/W responsivity at 1310 nm wavelength, which corresponds to a quantum efficiency of >94%[40] while AIM Photonics provides ones with quantum efficiency >80% at 1550 nm wavelength[41]. Moreover, there is more flexibility for the efficiency improvement if customised components can be used. Thus, a fully integrated version of our QRNG protocol is practically attainable with existing PIC technology.

At the system level, however, the main limitation of our current work is the relatively low randomness generation/expansion rate. We now discuss some reasons for the limited randomness expansion rate. One of the reasons is that we used Eve's guessing probability to construct the min-tradeoff function when applying the EAT. While the guessing probability can be easily computed using the SDP relaxation, this construction is generally not tight. Recently, a tight bound on the conditional von Neumann entropy that is also compatible with our SDP relaxation was developed[42]. It would be interesting to see if one can

obtain an improvement in the randomness expansion rate by incorporating the new bound on the conditional von Neumann entropy.

Another reason for the limited randomness expansion rate is that we coarse-grained the homodyne detection output. Indeed, here we considered a measurement device that gives a binary output; whereas in practice, the homodyne measurement actually has a large number of discrete bins determined by the ADC circuit. The coarse-graining was done to simplify the protocol as it allows us to simply monitor the winning probability in the parameter estimation step. However, our security analysis can be extended to consider more outputs using the framework of ref. [30]. Nevertheless, there is a tradeoff between the number of outputs used in the protocol and the randomness generation rate as fine-graining the outputs would enhance the effect of the statistical fluctuations as the probability of obtaining each bin gets smaller. Finding the optimal number of bins for the measurement outputs deserve a deeper investigation in the future.

Next, our protocol also demands relatively high detection efficiency to generate net randomness expansion. As there is a tradeoff between the electronic noise (and its equivalent efficiency loss) and the working bandwidth of the system, the working frequency in our experiment is relatively low which results in the low randomness generation rate. One of the reasons for the high detection efficiency requirement is the choice of states used in the protocol. In our experiment, we use the QPSK encoding for the simplicity of the experiment. However, this choice of states may be sub-optimal. For example, as shown in Fig. 3b, a significant increase in the randomness expansion per pulse and decrease in the required detection efficiency can be seen by choosing the QAM-16 constellation. This would lead to a significant increase in the actual randomness expansion rate.

Finally, in this work, we assume that the device is sufficiently isolated from the environment such that any quantum side information that is accessible to the adversary was obtained before the start of the protocol. While this assumption can be well justified for QRNGs as both parties are inside in the same secure location, it could still be removed using the recently developed generalisation of the EAT[43,44]. The relaxed assumption may be relevant in a more pessimistic scenario where the device is surrounded by an insecure environment such that any photons that are scattered in the channel might fall into the adversary's hands. Since the bound in the generalised EAT is similar to the one derived under the original entropy accumulation framework (with slightly different correction terms), we expect that our QRNG could still exhibit randomness expansion in the more pessimistic scenario. We leave this investigation for future work.

**Table 3 | The configuration for the P&M game $\mathcal{G}$ used in the test rounds in our experiment**

| System Setting | $x = 0$ $y = 0$ | $x = 0$ $y = 1$ | $x = 1$ $y = 0$ | $x = 1$ $y = 1$ | $x = 2$ $y = 0$ | $x = 2$ $y = 1$ | $x = 3$ $y = 0$ | $x = 3$ $y = 1$ |
|---|---|---|---|---|---|---|---|---|
| $q(x, y)$ | 0 | 0.256 | 0 | 0.232 | 0.244 | 0.012 | 0.244 | 0.012 |
| Score for $b = 0$ | 0 | 1 | 0 | 0 | 0 | 0 | 1 | 0 |
| Score for $b = 1$ | 1 | 0 | 1 | 1 | 1 | 1 | 0 | 1 |

The first row represents the probability of choosing a specific setting for Alice's input $x$ and Bob's input $y$. The second row and third row show the score assignment for each input-output configuration.

**Table 4 | Observed experimental results for the test rounds**

| System Setting | $x=0$ $y=0$ | $x=0$ $y=1$ | $x=1$ $y=0$ | $x=1$ $y=1$ | $x=2$ $y=0$ | $x=2$ $y=1$ | $x=3$ $y=0$ | $x=3$ $y=1$ |
|---|---|---|---|---|---|---|---|---|
| $P(b=0|x,y)$ | 0.4994 | 0.5952 | 0.4994 | 0.4023 | 0.4034 | 0.5022 | 0.5979 | 0.5006 |
| $P(b=1|x,y)$ | 0.5006 | 0.4048 | 0.5006 | 0.5977 | 0.5966 | 0.4978 | 0.4021 | 0.4994 |

To summarise, we present a QRNG protocol based on a completely uncharacterised homodyne detector. The security analysis takes into account the finite size effects and the non-i.i.d. measurement process, providing random number generation that is certified in the presence of quantum side information. To verify the feasibility of the protocol, we set up a high-efficiency and low noise fibre-coupled homodyne detector for experiment. The averaged quantum efficiency of the photodiode pair is 98.55% at 1550 nm, and the clearance is measured to be 16.94 dB with a 10 mW LO input. The effective efficiency of the homodyne detector is characterised to be 91.7%. In order to have a proper implementation of our protocol and remove potential signal distortions, we come up with a complementary modulation scheme and adopt the two-mode coherent states for quantum state preparation. This guarantees both the modulation signals and measurement outcomes are DC-balanced data streams, for any experimental setting during the QRNG protocol execution. The system works at a repetition rate of 2.5 MHz, and finally obtain a gross randomness generation rate of 0.00455 and a net randomness expansion rate of 0.00199, with a $1 \times 10^{10}$ rounds of protocol execution. In addition, we show that our protocol is compatible with the silicon photonics platform and is readily implementable on silicon PIC.

In conclusion, our results exhibit a practical QRNG with self-testing feature and provable security, showing a great potential for providing certifiable randomness for practical and private use.

## Methods
### Randomness certification

Before we explain the randomness certification of the protocol, we shall first explain the security criterion of the protocol. Consider a randomness generation protocol that produces an output string, which we label as **Z**. In a self-testing protocol such as ours, it is common that the legitimate parties exchange some classical information during the protocol (e.g., in the parameter estimation step). We denote the transcript of any classical communication in the protocol by **M**. Suppose also that the protocol involves seeded randomness extraction. We shall denote the seed used for the randomness extraction by **S**. Finally, any side information that is available to Eve will be denoted by $E$. For a given run of the protocol, let us suppose that its output can be described using the quantum state

$$\rho_{\mathbf{ZSM}E} = |\varnothing\rangle\langle\varnothing|_{\mathbf{Z}} \otimes \tau_{|\mathbf{S}|} \otimes \tilde{\rho}_{\mathbf{M}E}^{\varnothing} + \tilde{\sigma}_{\mathbf{ZSM}E}. \tag{4}$$

Here, we account for the probability that the protocol may abort (in which case, we denote the output of the protocol by $\varnothing$). Furthermore, $\tau_l$ denotes the uniformly random string with length $l$ denoted in the subscript and $\tilde{\rho}_{\mathbf{M}E}^{\varnothing}$ describes the sub-normalised state of Eve's side information and the classical transcript when the protocol aborts. On the other hand, the sub-normalised state $\tilde{\sigma}_{\mathbf{ZSM}E}$ in the second term describes the state when the protocol is not aborted

$$\tilde{\sigma}_{\mathbf{ZSM}E} = \sum_{z,s} |z,s\rangle\langle z,s|_{\mathbf{ZS}} \otimes \tilde{\rho}_{\mathbf{M}E}^{z,s}, \tag{5}$$

where the summation is taken over all possible output and seed strings, which we denote by $z$ and $s$. The sub-normalised state $\tilde{\rho}_{\mathbf{M}E}^{z,s}$ is the state describing Eve's side information and the classical transcript conditioned on the output string being $z$ and the seed string being $s$.

Denoting the event in which the protocol is not aborted by $\Omega$, we have

$$\begin{aligned}
\text{Tr}[\tilde{\rho}_{\mathbf{M}E}^{\varnothing}] &= 1 - \Pr[\Omega], \\
\text{Tr}[\tilde{\rho}_{\mathbf{M}E}^{z,s}] &= \Pr[\mathbf{Z}=z, \mathbf{S}=s], \\
\text{Tr}[\tilde{\sigma}_{\mathbf{ZSM}E}] &= \sum_{z,s} \Pr[\mathbf{Z}=z, \mathbf{S}=s] = \Pr[\Omega].
\end{aligned} \tag{6}$$

Normalising the state in which the protocol is not aborted, we obtain $\sigma_{\mathbf{ZSM}E} := \tilde{\sigma}_{\mathbf{ZSM}E} / \Pr[\Omega]$. We say that the QRNG is $\varepsilon_{\text{sou}}$-sound if

$$\Pr[\Omega] \cdot \frac{1}{2} \left\| \sigma_{\mathbf{ZSM}E} - \tau_\ell \otimes \tau_{|\mathbf{S}|} \otimes \sigma_{\mathbf{M}E} \right\|_1 \leq \varepsilon_{\text{sou}} \tag{7}$$

for a fixed $\varepsilon_{\text{sou}} \in (0,1)$. Here, $\sigma_{\mathbf{M}E} = \text{Tr}_{\mathbf{ZS}}[\sigma_{\mathbf{ZSM}E}]$. Informally speaking, the soundness of the protocol would imply that either the protocol aborts with high probability or the output of the protocol would be close (in trace-distance) to a random string with length $\ell$ that is independent of the seed **S**, any classical information being exchanged in the protocol **M**, and Eve's side information $E$. Importantly, the above security definition is composable. Hence, the output of the protocol can be securely used for other cryptographic applications.

However, a protocol that always aborts would trivially satisfy the soundness condition given in Eq. (7), and such a protocol is clearly undesirable. Therefore, we also impose an additional requirement that the protocol would succeed with high probability of producing a random string when the device works as expected. Formally, we call a protocol $\varepsilon_{\text{com}}$-complete if its honest implementation (which may use imperfect devices) satisfy the following

$$\Pr[\Omega]_{\text{honest}} \geq 1 - \varepsilon_{\text{com}} \tag{8}$$

for some fixed $\varepsilon_{\text{com}} \in (0,1)$. Note that the subscript "honest" emphasises that $\Pr[\Omega]_{\text{honest}}$ is calculated with the assumption that the device works as expected, in particular, independently and identically for each round. In this case, we normally model the behaviour of the device, including its imperfection, and calculate the probability of the protocol aborting (e.g., due to statistical fluctuations in the parameter estimation) in such scenario.

Next, to analyse the security of the protocol, we shall assume the following:
1. Quantum theory is correct.
2. Alice has a trusted source of quantum states that can accurately prepare the code states specified by the protocol.
3. The device is equipped with trusted and private random seed.
4. The device has access to trusted classical devices to perform any classical post-processing.
5. The device is well isolated such that it does not leak additional quantum side information nor the output string.

Now, we shall briefly elaborate the assumptions mentioned above. The first assumption is normally taken for granted as quantum theory is the best available description of nature at small scale that we currently have. As such, throughout this paper, we shall assume that Eve and the devices used in the protocol obey the laws of quantum physics.

The second assumption can be practically justified by careful characterisation of Alice's source. As the scenario that is relevant for QRNG considers the case where Alice and Bob are located in close proximity to each other, one could reasonably believe that the source

is well protected from source side-channel attacks that are possible in other quantum crytographic protocols (for example, Trojan horse attacks in QKD[45]). In particular, we assume that the source behaves identically and independently in each round.

The third assumption is necessary because the measurement device which generates the raw random string in this protocol is uncharacterised. If the inputs are not chosen from a trusted random number generator, one possible scenario is that the uncharacterised measurement device could have access to the inputs before the protocol is run. In this case, it is trivial to reproduce the statistics obtained by the honest implementation of the protocol. Moreover, as our randomness certification would utilise the EAT, using a trusted random number generator could enforce the quantum Markov chain condition. Lastly, the last step of the protocol uses seeded extraction, which requires a private and uniformly random seed.

The fourth assumption is necessary for any QRNG protocol to prevent the security criteria from being trivially broken. For example, when the randomness extraction is not executed properly, it is clear that the soundness criterion may not be satisfied. Furthermore, when the output string is leaked, it is trivial to guess the output of the QRNG.

The last assumption is necessary for two reasons. Firstly, it is obvious that the security of the protocol is null when the device leaks the output string. Secondly, due to some technicality with the EAT, we need to assume that the quantum side information available to Eve is not updated as the protocol is run. It is worth noting that this assumption is not too restrictive for QRNGs since Alice and Bob are both inside the same secure location. Recently, there is a generalised version of the EAT[43,44] that allows Eve's quantum side information to be updated as the protocol is run and hence, it would be interesting to see if the assumption that the device does not leak additional quantum side information can be relaxed.

Having mentioned the assumptions we need in the security analysis, we emphasise again that we do not make any assumptions on the measurement device and the quantum channel. In particular, in a given round, the behaviour of these components can have arbitrary correlation to their inputs and outputs in the preceding rounds (unlike the source which we assumed to behave independently and identically in each round). Remarkably, our protocol remains secure even if there is a degradation in the homodyne detector or when Eve has some pre-shared entanglement with Bob's uncharacterised measurement device.

As elaborated previously, we want our protocol to satisfy both soundness and completeness criteria. We first prove the completeness of our QRNG protocol. To that end, we use the following theorem.

**Theorem 2.** (Bounds on the binomial cumulative distribution[46,47]) Let $n \in \mathbb{N}$, $p \in (0, 1)$ and let $X$ be a random variable distributed according to $X \sim \text{Binomial}(n, p)$. Then, for any integer $k$ such that $0 \leq k < n$, we have

$$F(n, p, k) \leq \Pr[X \leq k] \leq F(n, p, k + 1),\qquad(9)$$

where

$$D(q, p) = q \ln\left(\frac{q}{p}\right) + (1 - q) \ln\left(\frac{1 - q}{1 - p}\right)$$

$$\Phi(a) = \frac{1}{\sqrt{2\pi}} \int_{-\infty}^{a} \mathrm{d}x\, e^{-x^2/2}$$

$$F(n, p, k) = \Phi\left(\text{sign}\left(\frac{k}{n} - p\right)\sqrt{2nD\left(\frac{k}{n}, p\right)}\right)$$

Since the protocol is aborted if $|\{C_i : C_i = 0\}| > n\gamma(1 - \omega + \delta)$, we can apply Theorem 2, in the same way as in ref. [9], to get the following

upper bound on the probability of the protocol being aborted

$$\Pr\left[|\{C_i : C_i = 0\}| > n\gamma(1 - \omega + \delta)\right] \leq 1 - F(n, \gamma(1 - \omega), \lfloor n\gamma(1 - \omega + \delta)\rfloor).\qquad(10)$$

Hence, by choosing the completeness error as

$$\varepsilon_{\text{com}} = 1 - F(n, \gamma(1 - \omega), \lfloor n\gamma(1 - \omega + \delta)\rfloor),\qquad(11)$$

our QRNG protocol would satisfy the completeness condition.

Next, to prove the soundness of our QRNG protocol, we shall use the following Quantum Leftover Hash Lemma (Theorem 8 of ref. [33]).

**Theorem 3.** (Quantum Leftover Hash Lemma[33]) Let $\rho_{\mathbf{B}E'}$ be a classical-quantum state and $\mathcal{F} = \{f_s : \{0, 1\}^n \to \{0, 1\}^\ell\}$ be a two-universal hash family with $\mathbf{Z} = f_s(\mathbf{B})$ and the seed $\mathbf{S} \in \{0, 1\}^m$ is chosen uniformly. Let $0 < \kappa \leq \varepsilon/2 < 1$, we have

$$\frac{1}{2}\left\|\rho_{\mathbf{Z}\mathbf{S}E'} - \tau_\ell \otimes \tau_m \otimes \rho_{E'}\right\|_1 \leq 2\left(\frac{\varepsilon}{2} - \kappa\right) + 2^{3/2}\left(2^{\ell - H_{\min}^{\varepsilon/2 - \kappa}(\mathbf{B}|E')}\right)^{1/4}\qquad(12)$$

where $\tau_\ell$ and $\tau_m$ are the uniform random strings of length $\ell$ and $m$ respectively. Consequently, if we choose the output length to be

$$\ell = \max_\kappa\left\lfloor H_{\min}^{\varepsilon/2 - \kappa}(\mathbf{B}|E') + 4\log_2\kappa - 2\right\rfloor,\qquad(13)$$

where the maximisation is taken over $\kappa \in (0, \varepsilon/2]$, then, we have

$$\frac{1}{2}\left\|\rho_{\mathbf{Z}\mathbf{S}E'} - \tau_\ell \otimes \tau_m \otimes \rho_{E'}\right\|_1 \leq \varepsilon.$$

On the other hand, for a fixed smoothing parameter $\epsilon_s$, the EAT (Theorem 1) guarantees that either the protocol aborts with probability of at least $1 - \epsilon_{\text{EA}}$ (i.e. the probability that the protocol is not aborted is upper bounded by $\epsilon_{\text{EA}}$) or the conditional smooth min-entropy $H_{\min}^{\epsilon_s}(\mathbf{B}|\mathbf{M}, E)_{\rho^\Omega}$ (here, $\mathbf{M}$ consists of the registers $\mathbf{T}$, $\mathbf{X}$, and $\mathbf{Y}$) is lower bounded by a certain amount. By identifying the register $E'$ in Theorem 3 as the register $\mathbf{M}E$ in the soundness criterion, we can choose the soundness error $\varepsilon_{\text{sou}}$ to be

$$\varepsilon_{\text{sou}} = \max\{\epsilon_{\text{EA}}, 2(\epsilon_s + \kappa)\}.\qquad(14)$$

In this case, EAT either upper bounds $\Pr[\Omega]$ by $\epsilon_{\text{EA}}$ or—in conjunction with the Quantum Leftover Hash Lemma—guarantees that the trace-distance term (for the state in which the protocol is not aborted) in the soundness criteria is smaller than $2(\epsilon_s + \kappa)$. Hence, our choice of $\varepsilon_{\text{sou}}$ ensures that the protocol is sound in both cases considered by the EAT. In this work, we choose $\varepsilon_{\text{sou}} = \epsilon_{\text{EA}} = 2(\epsilon_s + \kappa)$ where $\kappa$ is chosen to maximise the expected net expansion rate.

We shall now discuss the technical details of Theorem 1. Firstly, to apply the EAT, it is important to ensure that the so-called Markov condition is satisfied during the execution of the protocol. More precisely, we want that for any round $i \in [n]$, we need

$$I(B_{[i]} : X_{i+1}Y_{i+1}T_{i+1}|X_{[i]}, Y_{[i]}, T_{[i]}, E) = 0\qquad(15)$$

where $I(A : B|C)$ denotes the quantum mutual information between $A$ and $B$ conditioned on $C$. Here, $B_{[i]}$ denotes the string $(B_1, B_2, \ldots B_i)$ that describes the measurement outcomes from the first round until the $i$-th round. $X_{[i]}, Y_{[i]}, T_{[i]}$ are defined similarly. To enforce the Markov condition in the protocol, we implement each round sequentially and we choose the inputs for each round from a trusted and private random seed which is independent from the inputs and outputs from the preceding rounds. We also isolate the device such that Eve does not

obtain additional quantum side information as we execute the protocol.

The next ingredient we need is the so-called min-tradeoff function $f$ (for its formal definition, we refer the readers to Definition II.4 of ref. [30]). The min-tradeoff function is, roughly speaking, an affine lower bound on the worst case single-round conditional von Neumann entropy $H(B_i|X_i, Y_i, T_i, R)$ that is "compatible" with the probability distribution over $\mathcal{C} = \{\perp, 0, 1\}$ for random variable $C_i$. Here, $R$ is a quantum register that is isomorphic to the pre-measured state in round $i$.

To construct the min-tradeoff function, we follow the framework presented in ref. [30] in the context of device-independent randomness expansion. A key difference here is that we use the bound on the conditional von Neumann entropy derived in the Theorem 14 of ref. [48]

$$H(B_i|X_i = 0, Y_i = 0, T_i = 0, R) \geq 2\Big[1 - p_g(B_i|X_i = 0, Y_i = 0, T_i = 0, R)\Big] \tag{16}$$

instead of the bound based on conditional min-entropy used in ref. [30]. While both bounds are based on the guessing probability $p_g(B_i|X_i = 0, Y_i = 0, T_i = 0, R)$, the bound that we used here is significantly tighter than the one given by conditional min-entropy in the parameter regime in which the experiment is conducted. Another advantage is that the bound that we use is already linear, and as such, we do not need to perform the linearisation that was performed in ref. [30] to obtain an affine min-tradeoff function.

To bound the guessing probability, we use the semi-definite programming (SDP) technique proposed in ref. [31] instead of the Navascues–Pironio–Acin (NPA) hierarchy[49,50] used in ref. [30]. Both techniques are two similar hierarchies of SDP relaxation that bound the set of quantum correlations; the latter is for the device-independent scenario while the former is appropriate for the prepare-and-measure architecture considered in our protocol. For a fixed level of relaxation $k$, a given P&M game $\mathcal{G}$, characterised by the scoring coefficients $w_{b,x,y} = q(x,y)\delta_{b,b_{xy}}$, and some winning probability $v$, the SDP for the guessing probability has the following primal form

$$\begin{aligned}\max_{\{M_{b|y}\}_{b,y}, \{\Pi_e\}_e, \mathcal{U}} \quad & \sum_{b=0}^{1} \langle\phi_0|M_{b|0}\Pi_b|\phi_0\rangle \\ \text{subject to} \quad & \sum_{b,x,y} w_{b,x,y}\langle\phi_x|M_{b|y}|\phi_x\rangle = v, \\ & \langle\phi_x|\phi_{x'}\rangle = \langle\psi_x|\psi_{x'}\rangle \quad \forall x, x' \in \mathcal{X}.\end{aligned} \tag{17}$$

Here, $\{M_{b|y}\}_{b,y}$ denotes Bob's POVM elements, $\{\Pi_e\}_e$ denotes the POVM elements acting on the system $R$ and $\mathcal{U} : |\psi_x\rangle \to |\phi_x\rangle$ denotes the isometry describing the unknown quantum channel connecting Alice and Bob. From the dual solution of (17), we can obtain a bound on the guessing probability of the form

$$p_g(B_i|X_i = 0, Y_i = 0, T_i = 0, R) \leq c_v + \boldsymbol{\lambda}_v \cdot \boldsymbol{p}. \tag{18}$$

Here, $\boldsymbol{p} = (1 - p, p)$ is the score distribution of the device while $\boldsymbol{\lambda}_v$ and $c_v$ are the dual solutions to the SDP (17). We emphasise that $v$ is a parameter that we can choose freely and it does not have to be the actual winning probability that is attained by the device.

Following the arguments in ref. [30], consider the affine function $g_v$, which maps a distribution over $\mathcal{C} \setminus \{\perp\}$ to a real number

$$g_v(\boldsymbol{e}_c) = 2(1 - \gamma)\big[1 - c_v - \boldsymbol{\lambda}_v \cdot \boldsymbol{e}_c\big], \tag{19}$$

where $\boldsymbol{e}_c$ is the probability distribution where its $c$th entry is 1 and the other entries are zeros. Then, for some constant $u_\perp$ that we shall

determine later, the following function $f_v$ is a min-tradeoff function

$$\begin{aligned}f_v(\boldsymbol{e}_c) &= \frac{g_v(\boldsymbol{e}_c)}{\gamma} + \left(1 - \frac{1}{\gamma}\right)u_\perp, \quad \forall c \neq \perp \\ f_v(\boldsymbol{e}_\perp) &= u_\perp.\end{aligned} \tag{20}$$

To find the worst case over all distributions which lead to the protocol being accepted, we repeat the argument presented in ref. [9] here. First, we use the condition for the protocol to be accepted, $\text{freq}_{\mathbf{C}}(0) \leq \gamma(1 - \omega + \delta)$, to deduce that if $u_\perp \geq g_v(\boldsymbol{e}_0)$, then we have

$$f_v(\text{freq}_{\mathbf{C}}) \geq (1 - \omega + \delta)\big(g_v(\boldsymbol{e}_0) - u_\perp\big) + \frac{\text{freq}_{\mathbf{C}}(1)}{\gamma}\big(g_v(\boldsymbol{e}_1) - u_\perp\big) + u_\perp. \tag{21}$$

Now, we demand that $u_\perp \leq g_v(\boldsymbol{e}_1)$ and hence, the second term on the right-hand side can be dropped

$$f_v(\text{freq}_{\mathbf{C}}) \geq (1 - \omega + \delta)\big(g_v(\boldsymbol{e}_0) - u_\perp\big) + u_\perp. \tag{22}$$

This is increasing with $u_\perp$ and hence, it is best to fix $u_\perp = g_v(\boldsymbol{e}_1)$. We have

$$f_v(\text{freq}_{\mathbf{C}}) \geq (1 - \omega + \delta)g_v(\boldsymbol{e}_0) + (\omega - \delta)g_v(\boldsymbol{e}_1) \tag{23}$$

$$= 2(1 - \gamma)\big[1 - c_v - \boldsymbol{\lambda}_v \cdot \tilde{\boldsymbol{\omega}}\big], \tag{24}$$

where the adjusted score $\tilde{\boldsymbol{\omega}} = (1 - \omega + \delta, \omega - \delta)$. Thus, we have obtained the function $f$ in Theorem 1.

Lastly, we have to calculate the correction terms $V$ and $K$. To that end, we need to consider a few properties of the min-tradeoff function. They are the following

1. Maximum over all probability distributions

$$\text{Max}[f_v] = \max_{p \in \mathcal{P}_\mathcal{C}} f_v(p), \tag{25}$$

where $\mathcal{P}_\mathcal{C}$ is the set of all valid probability distributions.

2. Minimum over all protocol respecting distributions

$$\text{Min}_\Gamma[f_v] = \inf_{p \in \Gamma} f_v(p), \tag{26}$$

where $\Gamma$ denotes the set of distributions of the form $(\gamma\boldsymbol{\omega}, 1 - \gamma)$. We call such distribution a protocol respecting distribution.

3. The maximum variance over all protocol respecting distributions

$$\text{Var}_\Gamma[f_v] = \max_{p \in \Gamma} \sum_{c \in \mathcal{C}} p(c)[f(\boldsymbol{e}_c) - f_v(p)]^2. \tag{27}$$

To compute these quantities, we consider the maximum and minimum attainable value of $g_v$ (over all distributions) as

$$\begin{aligned}\text{Max}[g_v] &= 2(1 - \gamma)\big[1 - c_v - \lambda_{\min}\big], \\ \text{Min}[g_v] &= 2(1 - \gamma)\big[1 - c_v - \lambda_{\max}\big],\end{aligned} \tag{28}$$

where $\lambda_{\min} = \min \boldsymbol{\lambda}_v$ and $\lambda_{\max} = \max \boldsymbol{\lambda}_v$. Note that the choice of $g_v(\boldsymbol{e}_0) \leq u_\perp = g_v(\boldsymbol{e}_1)$ implies that $u_\perp = \text{Max}[g_v]$. Therefore, we are dealing with similar min-tradeoff functions as those considered in refs. [29,30], where we have the following relations

$$\begin{aligned}\text{Max}[f_v] &= \text{Max}[g_v], \\ \text{Min}_\Gamma[f_v] &\geq \text{Min}[g_v], \\ \text{Var}_\Gamma[f_v] &\leq \frac{(\text{Max}[g_v] - \text{Min}[g_v])^2}{\gamma}.\end{aligned} \tag{29}$$

Based on the above relations, we can calculate the correction terms $V$ and $K$. Following ref. [30], the correction term $V(\gamma, f_\nu)$ is given by

$$V(\gamma, f_\nu) = \frac{\ln 2}{2}\left(\log_2 9 + \sqrt{\frac{4(1-\gamma)^2(\lambda_{\max}-\lambda_{\min})^2}{\gamma}+2}\right)^2. \qquad (30)$$

On the other hand, the other correction term $K(\beta, \gamma, f_\nu)$ is given by

$$K(\beta, \gamma, f_\nu) = \frac{2^{\beta[1+2(1-\gamma)(\lambda_{\max}-\lambda_{\min})]}}{6(1-\beta)^3 \ln 2}\ln^3\left(2^{1+2(1-\gamma)(\lambda_{\max}-\lambda_{\min})}+e^2\right). \qquad (31)$$

Having specified the functions $f_\nu$, $V$ and $K$, we can now apply Theorem 1 and Theorem 3 to prove the soundness of our QRNG protocol. This concludes the randomness certification of the protocol.

## Input randomness

In the previous section, we have shown that if the extracted length $\ell$ is chosen according to Eq. (13), our QRNG protocol can generate randomness securely. However, for our QRNG protocol to be practically useful, we may also demand that, on average, it produces more randomness than the one consumed to run the protocol.

In this work, as we consider strong extractors for the randomness extraction, it is sufficient to consider the randomness consumed to choose the inputs **T**, **X** and **Y** as we can treat the extractor seed as part of the output. As the optimal input distribution is biased, one could either use a biased random seed or convert a uniform random seed into a biased one (for example, using the interval algorithm[51]). We denote the *expected* length of random bit string used to generate the inputs by $\ell_{\text{in}}$. The expected input randomness $\ell_{\text{in}}$ is approximately the Shannon entropy of the inputs (up to some small overhead that is negligible for large block sizes)

$$\begin{aligned}\ell_{\text{in}} &= H(\mathbf{T},\mathbf{X},\mathbf{Y}) + 3 \\ &= n[h_2(\gamma) + \gamma H(\boldsymbol{q})] + 3,\end{aligned} \qquad (32)$$

where we have used the fact that the inputs for each round are chosen independently from the ones from the preceding rounds and we also used the chain rule for Shannon entropy. Here, $h_2(\gamma)$ is the binary entropy function and $H(\boldsymbol{q})$ is the Shannon entropy of the input distribution $\{q(x,y)\}_{x,y}$.

## Homodyne detector modelling and characterisation

We model the homodyne detector from two aspects: optical loss and the electronic noise.

The optical loss arises from two main parts. The insertion loss of the BS, and the imbalance of the homodyne detection, which is caused by the efficiency mismatch of the two photodiodes and the imperfect BS splitting ratio.

We first characterise the photon detection efficiency of the photodiodes, which includes the quantum efficiency of the photodiodes, coupling loss to the photodiodes, and the insertion loss of the fibre-pigtailed GRIN lenses, with an optical power metre (EXFO PM-1100) and a Source Measurement Unit (Keysight U2722A).

The GRIN lenses used are anti-reflection-coated in the range of 1250–1650 nm, with an average reflection of <0.2%. In addition, the waist diameter of the output light beam is in the order of 10 μm, which is much smaller than the diameter of the active region of our PD (100 μm). We measure the detection efficiency of the two photodiodes, including the coupling loss and the insertion loss, by putting a reverse bias at the working voltage of the photodiode, giving a constant power input light from the laser, and measuring the photocurrent by the Source Measurement Unit. The efficiency of the two photodiodes is deduced from the ratio of the measured photocurrent and the input power to be 98.3% and 98.8%, respectively.

The splitting ratio of the beam splitter is measured to be 50.4:49.6, and the insertion loss is 0.2 dB. By matching the beam splitter with the PDs, and carefully balancing the amplitudes of the two arms, we gradually increase the input LO power with a variable optical attenuator (Yokogawa AQ2200-311A) and obtain the noise measurements. For frequency domain measurement, a spectrum analyser (Rohde & Schwarz FSV40) is used, with a resolution bandwidth of 1 MHz and a video bandwidth of 5 MHz. For the measurement of the noise variance and clearance, an oscilloscope (Tektronix MSO64 BW 2.5 GHz) is utilised.

The characterisation results are shown in Fig. 1d, e. With a 10 mW LO input, a clearance of 16.94 dB is obtained. Under the assumption that the electronic noise of homodyne detector possesses a Gaussian distribution and is independent of the measured optical signal, we follow the model proposed in ref. [52] and treat the effect of the electronic noise as equivalent to efficiency loss. In our case, an equivalent efficiency of 97.98% is estimated.

Taking all the factors into consideration, the total effective efficiency of our homodyne detector is characterised to be 91.7%.

## Two-mode coherent states

Two-mode coherent states are used in our system for quantum state preparation. Here, we give the basic form of the quadrature operator of the two-mode coherent state, and show that the quadrature value can be obtained by combining the quadrature values of individual temporal modes.

Without loss of generality, we define the creation (annihilation) operators of the early and late temporal modes by $\hat{a}_e^\dagger$ ($\hat{a}_e$) and $\hat{a}_l^\dagger$ ($\hat{a}_l$), respectively. Therefore, the quantum state composed of coherent states in both temporal modes can be expressed by:

$$\begin{aligned}|\alpha_e\rangle|\alpha_l\rangle &= e^{-\frac{|\alpha_e|^2}{2}}\sum_{m=0}^\infty \frac{(\alpha_e\hat{a}_e^\dagger)^m}{m!} \cdot e^{-\frac{|\alpha_l|^2}{2}}\sum_{n=0}^\infty \frac{(\alpha_l\hat{a}_l^\dagger)^n}{n!}|0\rangle \\ &= e^{-\frac{|\alpha_e|^2+|\alpha_l|^2}{2}}\sum_{k=0}^\infty \frac{(\alpha_e\hat{a}_e^\dagger+\alpha_l\hat{a}_l^\dagger)^k}{k!}|0\rangle \\ &= e^{-\frac{|\alpha_t|^2}{2}}\sum_{k=0}^\infty \frac{(\alpha_t\hat{a}_t^\dagger)^k}{k!}|0\rangle,\end{aligned} \qquad (33)$$

where $|\alpha_t| = \sqrt{|\alpha_e|^2+|\alpha_l|^2}$ and $\hat{a}_t^\dagger = \frac{1}{|\alpha_t|}(\alpha_e\hat{a}_e^\dagger+\alpha_l\hat{a}_l^\dagger)$ represent the amplitude and the creation operator of the new two-mode coherent state, respectively. In our case, we have $|\alpha_l\rangle = |-\alpha_e\rangle$, $|\alpha_t| = \sqrt{2}|\alpha|$, and $\hat{a}_t^\dagger = 1/\sqrt{2}(\hat{a}_e^\dagger - \hat{a}_l^\dagger)$.

As a result, the quadrature operator of the two-mode coherent states can be obtained (in Shot-Noise Unit):

$$\begin{aligned}\hat{q}_t &= \hat{a}_t e^{-i\theta} + \hat{a}_t^\dagger e^{i\theta} \\ &= \frac{\hat{a}_e - \hat{a}_l}{\sqrt{2}}e^{-i\theta} + \frac{\hat{a}_e^\dagger - \hat{a}_l^\dagger}{\sqrt{2}}e^{i\theta} \\ &= \frac{1}{\sqrt{2}}(\hat{q}_e - \hat{q}_l).\end{aligned} \qquad (34)$$

Hence, the quadrature value of the two-mode coherent state $q_t$ satisfies

$$\begin{aligned}\hat{q}_t|q\rangle &= \frac{1}{\sqrt{2}}(\hat{q}_e - \hat{q}_l)|q\rangle \\ &= \frac{1}{\sqrt{2}}(q_e - q_l)|q\rangle \\ &= q_t|q\rangle.\end{aligned} \qquad (35)$$

### Formulating a P&M game

Previously, we took for granted that our QRNG protocol specifies a P&M game that is used to test whether the devices are working as expected. However, constructing a game that is optimal for certifying randomness generation is a non-trivial task. In this section, we use the SDP duality to construct a P&M game that can asymptotically witness the same amount of randomness that is certified by full input-output probability distribution $\{P(b|x,y)\}_{b,x,y}$. The idea behind our method is to find a linear function of the input-output probability distribution that witnesses the randomness generated by Bob's measurement. Then, from this linear function, we could derive the input distribution $q(x,y)$ and the winning outputs $b_{xy}$. Similar constructions have been used in device-independent quantum information processing to construct an optimal Bell inequality for certifying randomness[53,54] and self-testing[55].

Asymptotically, we expect that the full input-output probability distribution should be optimal for witnessing randomness as it contains the full statistical information about the devices' behaviour. Let us now suppose that the expected input-output probability distribution is known (either by modelling the honest implementation or calibrating the device prior to the protocol). To construct a game that could optimally certify the amount of randomness, we shall consider the following SDP for the guessing probability subject to the full input-output probability distribution.

$$
\begin{aligned}
\max_{\{M_{b|y}\}_{b,y}, \{\Pi_e\}_e, \mathcal{U}} \quad & \sum_{b=0}^{1} \langle \phi_0 | M_{b|0} \Pi_b | \phi_0 \rangle \\
\text{subject to} \quad & \langle \phi_x | M_{b|y} | \phi_x \rangle = P(b|x,y), \forall b, x, y \\
& \langle \phi_x | \phi_{x'} \rangle = \langle \psi_x | \psi_{x'} \rangle, \quad \forall x, x' \in \mathcal{X}.
\end{aligned}
\tag{36}
$$

Suppose that the optimal dual solution to (36) for the $k$th level of relaxation is given by

$$
\hat{d}_k = \xi_0 + \sum_{b,x,y} \xi(b,x,y) P(b|x,y) \geq p_g(B_i | T_i = 0, X_i = 0, Y_i = 0, R),
\tag{37}
$$

where $\xi_0$ is associated with the non-statistical constraints. Any feasible dual solution is a linear function of the input-output distribution that upper bounds the guessing probability. As such, the set of feasible dual solutions to the SDP gives a family of linear upper bounds on the guessing probability while $\hat{d}_k$ is the tightest upper bound on the guessing probability among the family (in fact, it is "tight" up to the semi-definite relaxation[31] of the set of quantum correlations and any duality gap).

Now, for each pair of inputs $(x,y)$, we define $b'_{xy}$ and $b''_{xy}$ such that $\xi(b'_{xy}, x, y) \geq \xi(b''_{xy}, x, y)$. We consider

$$
\begin{aligned}
& \hat{d}_k - \xi_0 - \sum_{x,y} \xi(b''_{xy}, x, y) \\
& = \sum_{x,y} \Big[ \xi(b'_{xy}, x, y) P(b'_{xy}|x,y) + \xi(b''_{xy}, x, y) P(b''_{xy}|x,y) \\
& \quad - \xi(b''_{xy}, x, y) \big\{ P(b'_{xy}|x,y) + P(b''_{xy}|x,y) \big\} \Big] \\
& = \sum_{x,y} \big\{ \xi(b'_{xy}, x, y) - \xi(b''_{xy}, x, y) \big\} P(b'_{xy}|x,y),
\end{aligned}
\tag{38}
$$

where in the first equality we use the normalisation constraint. Noting that we are just subtracting a constant from $\hat{d}_k$, expression (38) is still an almost tight witness on the guessing probability. Finally, we could also divide the above expression by a constant and the resulting expression

$$
\sum_{x,y} q(x,y) P(b'_{xy}|x,y),
\tag{39}
$$

with

$$
q(x,y) = \frac{\xi(b'_{xy}, x, y) - \xi(b''_{xy}, x, y)}{\sum_{x,y} \big\{ \xi(b'_{xy}, x, y) - \xi(b''_{xy}, x, y) \big\}},
\tag{40}
$$

would still be an almost tight witness on the guessing probability. By construction, $\{q(x,y)\}_{x,y}$ is a valid probability distribution. Moreover, as $b'_{xy}$ is defined such that $\hat{d}_k$ (and hence, the bound on the guessing probability) would be higher when $P(b'_{xy}|x,y)$ increases, we could interpret $b'_{xy}$ as the losing outcome (i.e., we assign the score $C=0$) when the inputs $x$ and $y$ are chosen and $q(x,y)$ as the probability of choosing this pair of inputs. In this case, the game $\mathcal{G}$ is defined as one where for Alice and Bob choose the inputs $(x,y)$ with probability $q(x,y)$ and the winning outcome is given by $b_{xy} = b''_{xy}$.

The game construction that we have described above is almost optimal to witness the *generated* randomness. However, the construction may not minimise the randomness consumed to choose the inputs and hence, its performance in terms of the expanded randomness may be far from optimal. Formulating the optimal game construction that maximises the net randomness expansion rate would be an interesting direction for future work.

### Randomness extraction

Toeplitz hashing utilises a Toeplitz matrix, **H**. The Toeplitz matrix is an $m \times n$ diagonal-constant matrix, and it is constructed by filling up the first column and first row of the matrix with a uniform seed, denoted by **S**. Thus, the seed length required for Toeplitz hashing is $n + m - 1$ bits. Toeplitz matrix **H** can be expressed as

$$
\mathbf{H} = \begin{bmatrix}
s_n & s_{n-1} & \cdots & s_2 & s_1 \\
s_{n+1} & s_n & \cdots & s_3 & s_2 \\
\vdots & \vdots & \ddots & \vdots & \vdots \\
s_{n+m-1} & s_{n+m-2} & \cdots & s_{m+1} & s_m
\end{bmatrix}.
$$

The hashing is done by expressing the raw bits as a column vector and performing matrix-vector multiplication with the Toeplitz matrix. We used the calculated bit generation rate of 0.00455 and constructed a Toeplitz matrix **H** with the parameters $m = 45$ and $n = 10,000$.

A field programmable gate array (FPGA) is the chosen platform for implementation of Toeplitz hashing. The schematic of our post-processing on FPGA is shown in Fig. 5. The raw data is stored on a personal computer (PC) and sent in batches of approximately 600 Mbits via 1G Ethernet to the FPGA. Upon receiving the batch of raw

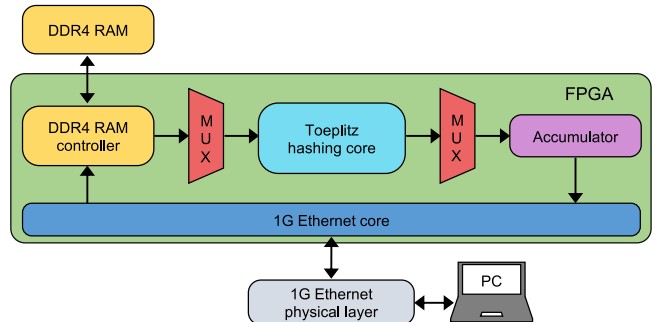

**Fig. 5 | Schematic of the post-processing on the ZCU111 evaluation board.** The raw numbers are stored on a personal computer (PC) and sent to the field programmable gate array (FPGA) via 1G Ethernet. The data is stored in the double data rate 4th generation random access memory (DDR4 RAM) on the processing system (PS) and multiplexed into the Toeplitz hashing core on the programmable logic (PL) side of the FPGA. The output from the Toeplitz hashing core is accumulated and sent back to PC via 1G Ethernet.

data, the processing system (PS) on FPGA sends in 10 kbits of data to the programmable logic (PL), where the data will be further split into 10 batches of 1 kbits each. The multiplexing of raw data and seed is done via pipelining and the Toeplitz hashing algorithm is executed in parallel. The PS then receives the output of 45 bits from PL, and sends in a new set of 10 kbits of data to PL, repeating until all the data in the current batch has been processed. The PS then sends the extracted random numbers of the current batch to PC via Ethernet and waits for a new batch, until all 10.622 Gbits of raw data has been processed.

## Data availability
All of the data that support the findings of this study are available in the main text. Source data are available from the corresponding author on request.

## Code availability
The codes used for simulation are available from the corresponding author on request.

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

## Acknowledgements

The authors acknowledge funding support from the National Research Foundation of Singapore (NRF) Fellowship grant (NRFF11-2019-0001) and NRF Quantum Engineering Programme 1.0 grant (QEP-P2). Charles Lim contributed to this work while employed by JP Morgan Chase & Co. This work is for information purposes only, and is not a product of the Research Department of JPMorgan Chase & Co, or its affiliates. Neither JPMorgan Chase & Co nor any of its affiliates make any explicit or implied representation or warranty and none of them accept any liability in connection with this work, including, but limited to, the completeness, accuracy, reliability of information contained herein and the potential legal, compliance, tax or accounting effects thereof. This document is not intended as investment research or investment advice, or a recommendation, offer or solicitation for the purchase or sale of any security, financial instrument, financial product or service, or to be used in any way for evaluating the merits of participating in any transaction.

## Author contributions

C.L., I.W.P. and C.W. designed the research. C.W. designed and carried out the main experiment. I.W.P. developed the security proof and did the numerical simulation. H.J.N. implemented the post-processing. J.Y.H. and R.H. contributed in high-efficiency homodyne detection. J.Z. and G.Z. provided discussions on photonic chip platform and conducted testing. C.W. and I.W.P. wrote the manuscript with contributions from all authors.

## Competing interests

The authors declare no competing interests.
