## [Peer Review File · Nature Communications]

Provably-secure quantum randomness expansion with uncharacterised homodyne detectionREVIEWER COMMENTS

Reviewer #1 (Remarks to the Author):

In the work 'Provably – secure quantum randomness expansion with uncharacterized homodyne detection' the authors present a new semi-device independent quantum random number generator (QRNG) protocol and they implement it in a prove of principle experiment.

QRNGs have been proven to be fertile terrain for new and interesting research in the quantum technology world, mainly in the last few years many 'semi-device independent protocol' have been presented. The main idea, as pointed out by the authors, is to have a QRNG where we can leave much of the implementation uncharacterized and still be able to prove its 'quantumness' (allow me this neologism). I feel that in the past few years many of these new protocols have been a little bit of a hit or miss, however in this case I pleasantly surprised to have found a very nice work.

First of all, the authors present a SDI-protocol where the part left uncharacterized (the detection scheme) is actually the most complicated and difficult to model (its hard to distinguish electronic noise and shot noise and prove which is which while running the experiment). ;oreover, the protocol takes into account the ammount of randomness needed at the input and proves a net expansion over it (many other approaches skip this important passage).

The protocol is new and uses different known techniques (such as the Entropy accumulation theorem, the Quantum Leftover Hash Lemma) and some developed previously by the same group (such as the SDP method of Ref. (55)) to prove its security against a quantum side information. Even if this considers only the case where the attacker got entangled with the measurement apparatus prior to the experiment running. The authors however justify the assumption and promises a possible generalization of it.

Moreover, the authors present a first implementation of the protocol using a laser, few modulators (intensity and phase) and a homodyne detection scheme. The experiment is complete and presents also the last step of randomness generation in these scenarios, consisting in the randomness extraction that, in this case, is performed on a field programmable gate array FPGA. The net randomness extraction rate is of around 5 KHz, which is arguably on the low side of QRNG, comparable to the randomness generation rates obtained by the more demanding Device Independent QRNGs (such as Y. Liu Nature volume 562, 548–551 (2018) rete of hundreds of bits/s). However, very differently from DI-QRNGs for this experiment there is a lot of margin for improvement (one way could be to implement the experiment in PIC as the authors suggests) and the experiment is considerably more practical.

All that said I conclude by saying that: I find the work very interesting and even if it appeals the most to a specialized public it is well written and includes both theory and experiment giving a broader audience the possibility to understand its content and appreciate its results to a full extent. Moreover, the novel approach presented here gives a considerable advancement in the field of SDI QRNGs giving security against quantum side information and keeping a practical and feasible implementation.

For these reasons I am inclined to suggest this paper for publication on the journal Nature Communication once the authors address few of my comments and observations.

- The authors, in Table I, presents a very informative resume of many different QRNG approaches giving a lot of useful information on the different protocol used. I was thinking if it was possible also to give some information about the performance of each approach. (Even if multiple works are bundle under one line, the best out of all of them could be used). Thereason why I suggest this is because I find this table very useful and having also the performance of the QRNG listed would make it even more effective for possible new protocols to be framed inside the existing literature (both in terms of assumptions and performances).

- Talking about the assumptions, I find very nice to have all the ones used for this work listed clearly with an explanation in the methods of this article. I have just one question about the second assumption. How accurately can the author characterize the quantum state prepared by Alice? How much would some imperfection in the state preparation affect the protocol if considered in the security proof?

- In the implementation everything is clear to me except the need for the dual pulse configuration. Is this done mainly for the homodyne detectors or for the modulators? If it is mainly for the

former, is not the output signal already balanced around zero? (since in the testing round a random phase is prepared and, in the generation round, signal and local oscillator have a $\pi/2$ phase difference). If it is mainly for the modulators, couldn't the compensation be done outside the time bin of the pulses? Maybe I am missing something but, in this way, it gives the impression to be an avoidable complication.

- I have another comment regarding the extraction. As far as I understand the authors send the data from a PC to a FPGA that extracts the data and sends it back. The authors also claim that "A FPGA is the chosen platform for implementation of Toeplitz hashing...". I would like to disagree. Usually it is useful to do an extraction directly on the FPGA itself because it is also used for every other digital processing in the experiment, however Toeplitz extraction would be much simpler in the case of a CPU or GPU. In fact, Toeplitz matrix multiplication can be easily converted in a convolution of two arrays of numbers, operation that can be efficiently done with the help of an FFT algorithm (e.g. in the article B.-Y. Tang et al. "High-speed and Large-scale Privacy Amplification scheme for Quantum Key Distribution" Scientific Reports 9 15733 (2019)). Since the data is already on a PC would not be simpler to extract it already there? I am not against the use on an FPGA but maybe the authors should motivate it differently.

- Just a minor remark about the presentation. I find the figure clear and informative, however the order in which they appear and the order in which they are referenced in the text do not match. For example, Fig. 1 is firstly referenced in page 7 (where it is explained) though it appears already in page 2. I think I would be clearer for the reader to have the experimental setup shown directly in the "Experimental implementation" section. Same goes for Fig. 3, Fig. 5 and Fig. 6. Also, (unless I miss it) I don't find Fig. 8 referenced anywhere in the text.

- Another small note is about the "Discussion" section, where the authors give an explanation about the feasibility of the experiment in a photonic integrated circuit. While I find this to be worthwhile to read, I find that it could be shortened and/or moved to the method section since I don't find the integrability of chip the main goal of this work but more an outlook that paves future development. Of course, this is just a suggestion and the authors are free to choose otherwise.

Reviewer #2 (Remarks to the Author):

This manuscript presents a semi-DI (more accurately source device independent) protocol for randomness generation/expansion. The works could be very interesting for the community, while there are a few points that need to be precisely addressed before I can give my final assessment of this manuscript:

1-This work requires the source to be characterized, which is a strong assumption compared to the other SDI protocols (Nat Commun 9, 5365 (2018), PRX 10, 041048 (2020), etc.); what are the most significant advantages of this protocol considering the rate is much lower than the similar works.

2- The setup is similar to the one presented in Refs. 24(No IID), 28 (with IID); indeed, quantum randomness can be extracted when the state's energy is bounded in the single photon energy level. With a similar approach, one can extract more randomness, as no game is needed, so the rate would be much higher. Is this approach considered only to consider quantum side-information rather than only classical one?

3- In "it is generally believed that the first practical application of such DI QRNGs will likely be Randomness Beacons", what exactly do the authors mean by "Randomness Beacons"? Citations 6 & 7 are DI QRNG and don't have any info on DI-QRNG in Randomness Beacons.

4-Concerning the experimental setup, as the homodyne detection is sensitive to phase, how do they manage to actively stabilize the phase.

5-In "In a P&M game, Alice receives a random input x from a pre-defined set $X \dots$ ", what are the requirements on the random input x ? should it be generated from a QRNG, or can it be generated

by any classical generator and possibly known by Eve?

6-It is claimed that "In particular, we do not assume that Bob's measurement device behaves independently and identically for each round. "what if there are some sorts of memory effects between each round accessible to Eve.

7-Considering there are inputs to the source and measurement (x,y) and the final entropy rate, is this protocol expanding the input randomness? Especially for commercially available detectors with much lower efficiency (~20%), I doubt the practicability of this proposal.

8-In "Furthermore, the honest implementation corresponds to homodyne detection... " what does "honest implementation" mean?

9-In Fig. 3, is the electronic noise constant regardless of LO?
Have they checked whether the photodiodes are saturated in high LO powers?

10- In Fig. 4, by system efficiency, does it mean detectors efficiency or efficiency of the whole design? Including the efficiency of components used in the source and measurement sides?

11-In Fig. 4, 10^{10} round experimental data is reported; how does it work, and is it averaged over all rounds?

12-Minor point in Fig. 4, there are discontinuities in the figure, which could be due to numerical precision, while it could be due to some minor issues in the optimization problem. Unfortunately, based on the security analysis represented in the paper, I couldn't reproduce the results given in Fig. 4, as the work should be reproducible; I was wondering if the authors have some guidance on the optimization problem or some repository with the codes?

13-Very minor point is the figures' positioning, which can be easily fixed.

14- Finally, even though the source is trusted in this protocol, still the rate is not significantly higher than fully DI QRNG; it would be great if the author could justify this point.

RESPONSE TO REVIEWERS' COMMENTS

Firstly, we would like to thank both reviewers for taking their time to carefully read our manuscript and give constructive feedback to our work. We are especially glad that both reviewers find our work very interesting. We have carefully considered their suggestions and made the appropriate modifications to our manuscript based on their valuable feedback. In the revised manuscript, we have re-arranged the Figures as suggested by the reviewers. As we did not make any significant changes in the main text, we do not attach any copy of the manuscript with highlighted changes.

RESPONSE TO REVIEWER 1

- 1. The authors, in Table I, present a very informative resume of many different QRNG approaches giving a lot of useful information on the different protocols used. I was thinking if it was possible also to give some information about the performance of each approach. (Even if multiple works are bundled under one line, the best out of all of them could be used). The reason why I suggest this is because I find this table very useful and having also the performance of the QRNG listed would make it even more effective for possible new protocols to be framed inside the existing literature (both in terms of assumptions and performances).**

We thank the reviewer for the suggestion. However, as the works compiled in Table 1 were done across different time periods (which would affect the specifications of the experimental apparatuses) and also different security definitions and/or parameters, we also think that there is a danger that readers may be misled by the reported performance (e.g., in terms of the randomness generation rate) of each approach. As a concrete example, the scheme proposed in Ref. [12] is virtually identical to the one proposed in Ref. [16] where the only difference is Ref. [12] analysed the security of the protocol against classical side-information while Ref. [16] analysed the security of the protocol against quantum side-information. As it turned out, the performance of Ref. [16] is superior to that of Ref. [12] (which is counter-intuitive because quantum side-information should give Eve significantly more power than classical side-information) but this is purely because Ref. [16] used experimental apparatus with better specifications. Furthermore, the earlier works on QRNG did not adopt the currently well-accepted composable security definition and hence, comparing their performance to more recent works would not be meaningful. Lastly, even among recent works which adopt the well-accepted security definition, there is no universally agreed security parameter using which the performance of QRNG should be measured. In this case, comparing randomness generation rates with different security parameters would not be fair.

For the reviewers' reference, we also attach the compiled randomness generation/expansion rate of each work together with this reply in a separate file.

- 2. Talking about the assumptions, I find it very nice to have all the ones used for this work listed clearly with an explanation in the methods of this article. I have just one question about the second assumption. How accurately can the author characterize the quantum state prepared by Alice? How much would some imperfection in the state preparation affect the protocol if considered in the security proof?**

Our formalism only demands the characterisation of the inner-products of the states and hence, it is versatile enough to allow for some uncertainty in the state preparation if we are able to characterise the deviation from the ideal states. For example, if we find that the states are mixed, we can purify the state and give its purification to Eve or a trusted shield system. As this is not the focus of the work, we aim to minimise this state preparation uncertainty and simply perform the security analysis assuming perfect state preparation.

In the experiment, we randomly modulate the phase of the coherent states to prepare the required quantum states. Therefore, the precision of the phase modulation determines the accuracy of the quantum state preparation. The average value of the phase deviations is measured to be 5.6×10^{-3} rad (0.32 deg), and the standard deviation of the phase modulation is measured to be 1.5×10^{-2} rad (0.86 deg).

- 3. In the implementation everything is clear to me except the need for the dual pulse configuration. Is this done mainly for the homodyne detectors or for the modulators? If it is mainly for the former, is not the output signal already balanced around zero? (since in the testing round a random phase is prepared and, in the generation round, signal and local oscillator have a $\pi/2$ phase difference). If it is mainly for the modulators, couldn't the compensation be done outside the time bin of the pulses? Maybe I am missing something but, in this way, it gives the impression to be an avoidable**

complication.

We thank the reviewer for the comment. The dual pulse configuration is for the DC balance of the output signal of the homodyne detector. Although quantum states are randomly prepared in the testing rounds, the probabilities of selecting each quantum state and measurement basis are subjected to the optimal P&M game that maximises the randomness generation rate, which may not be uniformly distributed (please refer to Table III for the optimal P&M game used in our experiment). As a result, the single pulse configuration cannot always guarantee a DC-balanced homodyne detection output signal with the optimal P&M game settings.

By adopting the two-mode coherent state configuration, the outputs of the homodyne detector are naturally DC-balanced for all experimental settings. This guarantees good signal integrity of the homodyne detector outputs, to get the highest randomness generation rate with the optimal P&M game settings.

4. **I have another comment regarding the extraction. As far as I understand the authors sends the data from a PC to a FPGA that extracts the data and sends it back. The authors also claim that “A FPGA is the chosen platform for implementation of Toeplitz hashing...”. I would like to disagree. Usually it is useful to do an extraction directly on the FPGA itself because it is also used for every other digital processing in the experiment, however Toeplitz extraction would be much simpler in the case of a CPU or GPU. In fact, Toeplitz matrix multiplication can be easily converted in a convolution of two arrays of numbers, operation that can be efficiently done with the help of an FFT algorithm (e.g. in the article B.-Y. Tang et al. “High-speed and Large-scale Privacy Amplification scheme for Quantum Key Distribution” Scientific Reports 9 15733 (2019)). Since the data is already on a PC would not be simpler to extract it already there? I am not against the use on an FPGA but maybe the authors should motivate it differently.**

Indeed, it would be simpler to directly perform the extraction on a PC, and it is also true that the FFT algorithm speeds up matrix multiplication. However, benefits from performing the extraction on FPGA include (1) FPGAs are much smaller in size and utilise much less resources than CPUs, (2) the power consumption of FPGA is significantly lower than CPU, and (3) doing so paves the way for future work where other components (e.g. control signals and analogue to digital converters) are included in the FPGA board as well. We thank the reviewer for the insightful comments, and have decided to remove the sentence to avoid confusing readers.

5. **Just a minor remark about the presentation. I find the figure clear and informative, however the order in which they appear and the order in which they are referenced in the text do not match. For example, Fig. 1 is firstly referenced in page 7 (where it is explained) though it appears already in page 2. I think I would be clearer for the reader to have the experimental setup shown directly in the “Experimental implementation” section. Same goes for Fig. 3, Fig. 5 and Fig. 6. Also, (unless I miss it) I don’t find Fig. 8 referenced anywhere in the text.**

We thank the reviewer for pointing this out. We have adjusted the ordering of the figures as suggested.

6. **Another small note is about the “Discussion” section, where the authors give an explanation about the feasibility of the experiment in a photonic integrated circuit. While I find this to be worthwhile to read, I find that I could be shorten and/or moved to the method section since I don’t find the integrability of chip the main goal of this work but more an outlook that paves future development. Of course, this is just a suggestion and the authors are free to choose otherwise.**

We thank the reviewer for the suggestion. Since this work aims at providing a practical QRNG scheme for private use, we still believe that it would be helpful and important to discuss the feasibility of our scheme to be implemented on PIC platforms. The discussion justifies that our scheme is promising for real-life QRNG applications with appealing features of cost-effectiveness and a small footprint.

RESPONSE TO REVIEWER 2

1. **This work requires the source to be characterized, which is a strong assumption compared to the other SDI protocols (Nat Commun 9, 5365 (2018), PRX 10, 041048 (2020), etc.); what are the most significant advantages of this protocol considering the rate is much lower than the similar works.**

We thank the reviewer for the comment. The reviewer correctly pointed out that our proposed QRNG requires the source to be (partially) characterised. However, we think that whether this assumption is stronger than the aforementioned works (which does not require the source to be characterised but requires the measurement device to be characterised) is debatable and depends on several factors, e.g., ease of implementation, device vulnerabilities, and availability of good security proofs.

To begin with, our security analysis only requires partial characterisation of the source in terms of the inner-product of the prepared states. For example, the dimension of the Hilbert space, which is impossible to verify in practice, is not required to be specified. In practice, however, the inner-product characterisation is only tight for pure states. To do this, we assume that the laser emits coherent states – a common assumption in quantum cryptography. The inner-product assumption can then be practically verified (e.g., by interfering the coherent states).

Next, we have argued against trusting the characterisation of the homodyne detector due to its complexity and hence, the difficulty in its modelling and characterisation. Also, from a hacking point of view, it also appears more difficult to ensure a trustworthy homodyne detection in practice. Recently, an out-of-band electromagnetic injection attack (Smith *et al*, *Phys. Rev. Applied* **15**, 044044 (2021)) that targets the hardware between the photodiodes and ADC has demonstrated the vulnerability of the measurement device in a continuous-variable QRNG. As the attack exploits an electromagnetic side-channel, Smith *et al* showed that it can be executed remotely and gain full control of the output whilst remaining undetected. Unlike the two protocols mentioned by the reviewer (as well as other QRNG protocols with trusted measurement devices), our protocol is completely immune to any detection side-channel attacks.

Admittedly, our protocol is also susceptible to side-channel attacks that manipulate the quantum state preparation. However, one can have countermeasures such as equipping the source with an isolator — the same cannot be said about the measurement device as it is meant to receive signals from the channel. Furthermore, based on the existing works on quantum hacking, it seems to suggest that the measurement device is the weakest link in quantum cryptographic protocols (although, to be fair, most of these works were done in the context of QKD but the work of Smith *et al* may seem to suggest that similar conclusion can be drawn for QRNG).

2. **The setup is similar to the one presented in Refs. 24 (No IID), 28 (with IID); indeed, quantum randomness can be extracted when the state’s energy is bounded in the single photon energy level. With a similar approach, one can extract more randomness, as no game is needed, so the rate would be much higher. Is this approach considered only to consider quantum side-information rather than only classical one?**

The reviewer correctly identified that one of the main reasons for the gap in the performance of our approach as compared to the works that quantum side-information is being considered here. In the worst case scenario, Eve may be entangled to the state being measured by Bob (Eve holds the purification of Bob’s state) and hence, she can guess Bob’s measurement outcomes better than what classical resources (such as shared randomness) allow her to. However, we do not think that formulating the testing of the protocol in terms of P&M games is the reason for the lower randomness generation rate as the P&M game was designed to certify the same randomness certified by the full statistics as explained in the Methods section.

3. **In "it is generally believed that the first practical application of such DI QRNGs will likely be Randomness Beacons", what exactly do the authors mean by "Randomness Beacons"? Citations 6 & 7 are DI QRNG and don’t have any info on DI-QRNG in Randomness Beacons.**

We thank the reviewer for the comment. In Ref. [7] (L. K. Shalm *et al*, *Nat. Phys.* **17**, 452 (2021)), the authors mentioned “Due to the size and complexity of a loophole-free Bell test, the first practical application of device-independent random number generators will likely be as a source of public randomness in randomness beacons.”

Randomness beacon produces timed outputs of fresh public randomness. It can be used for certain cryptography applications, public affairs that involve randomness like lottery, election auditing, etc. Some service provider of randomness beacons are listed below:

- (a) NIST: <https://beacon.nist.gov/home>

- (b) Random UChile: <https://random.uchile.cl>
- (c) Brazilian Beacon: <https://beacon.inmetro.gov.br>
- (d) League of Entropy: <https://www.cloudflare.com/leagueofentropy/>

4. Concerning the experimental setup, as the homodyne detection is sensitive to phase, how do they manage to actively stabilize the phase.

We thank the reviewer for the comment. Indeed, the relative phase between the signal and LO was actively stabilised during the experiment, since our fibre-optical system is quite sensitive to ambient environmental changes.

In order to keep track of the relative phase drift and actively stabilise it, we first put the fibre interferometer, including the signal path and the LO path, into a temperature-stabilised box. This helps to maintain the speed of the phase drift within a small range. Moreover, as illustrated in Fig. 6 in the manuscript, reference signals were sent periodically to characterise the relative phase drift between the two optical paths. Thereafter, a feedback control signal is added to the phase shifter (phase modulator for LO) to actively compensate the phase drift.

Additionally, we emphasise that the stability of the LO is mainly to achieve higher performance for the QRNG and the end users do not need to worry about this. The security of the random numbers is independent of the stability of the LO as the LO is part of the untrusted measurement device. Indeed, the difficulty to guarantee the stability of the LO is one of the motivations for treating the homodyne detector as an uncharacterised measurement.

5. In "In a P&M game, Alice receives a random input x from a pre-defined set X ...", what are the requirements on the random input x ? should it be generated from a QRNG, or can it be generated by any classical generator and possibly known by Eve?

Our security analysis demands that Eve's attacks (the action of the quantum channel and Bob's measurement device) are independent of x . So, to answer the question of the reviewer, the best practice is to generate the inputs from a QRNG to ensure the independence (although, what we truly require is the independence between the attack and the inputs). After the measurement outcome has been generated, x (in fact, Bob's inputs y as well) can be made known to Eve – in our security analysis, we certify randomness against Eve's quantum side information E and the inputs, x and y .

6. It is claimed that "In particular, we do not assume that Bob's measurement device behaves independently and identically for each round. "what if there are some sorts of memory effects between each round accessible to Eve?"

Before we discuss the possible memory effects that might be present in our QRNG, let us first define a "session" of the protocol and how it is different from a "round". A round is a period in which the parties receive inputs X , Y and T and the measurement device outputs the outcome B . A session of the QRNG protocol consists of running these rounds sequentially for n times and performing the classical post-processing as prescribed in Protocol 1.

The entropy accumulation theorem (EAT) allows any correlation (or memory effects) between different rounds (of the same session of the protocol) as long as the quantum Markov condition (Eq. (15)) is obeyed. In particular, a process where the output in the i -th round depends on inputs and outputs of the 1st round until the $(i-1)$ -th round obeys the Markov condition as long as the inputs for the $(i+1)$ -th round are chosen from a trusted RNG. We also require that the Gram matrix for the prepared states to be the same in all rounds. The last requirement is implicitly assumed within our trusted source assumption. Except for the IID assumption on the state preparation, we think that our model is sufficiently general to capture any memory effects that can happen between different rounds within the same session.

However, what our security analysis does not consider is the memory effects between different sessions of the protocol. As such, our protocol is susceptible to the same memory attack (Barrett *et al*, *Phys. Rev. Lett.* **110**, 010503 (2013)) that plagues device-independent quantum cryptography. There, the untrusted device may leak some private data (such as some part of the final random string) from the previous session by using the *public* communication channel that is used in the protocol. For QRNG, as Alice and Bob are located in the same secure location, we can assume that the protocol uses a *private* communication channel as a countermeasure to the memory attack. Note, however, that this does not constitute a security proof and it does not rule out a more sophisticated attack.

7. Considering there are inputs to the source and measurement (x,y) and the final entropy rate, is this protocol expanding the input randomness? Especially for commercially available detectors with much lower efficiency ($\sim 20\%$), I doubt the practicability of this proposal.

As shown in Fig. 4 (which plots the expected randomness expansion rate – *after* accounting for the randomness that we invested to generate the inputs – vs the system efficiency), our protocol can indeed expand randomness. Admittedly, as indicated by that figure – the protocol places a strict requirement on the quality of the device (we need the efficiency to be more than 80% and hence, we need a better detector efficiency than the one quoted by the reviewer) but we think that it is not beyond the reach of current technologies. Indeed, there are high efficiency photodiodes even on silicon PICs (as we mentioned in the Discussion section). While the economic aspect of our protocol is not the focus of this work, we believe a large scale production of the QRNG can bring cost down and hence, the protocol can still be practical. Furthermore, one could potentially improve the randomness generation rate of the protocol by also performing randomness extraction on the inputs (but this requires us to estimate the entropy of the inputs (X, Y, T) conditioned on Eve’s side information). In this case, the input randomness (X, Y, T) is not "wasted". We leave this improvement for future work.

8. In "Furthermore, the honest implementation corresponds to homodyne detection..." what does "honest implementation" mean?

In (semi) device-independent frameworks, the security analysis is agnostic to the implementation of the protocol. However, when assessing the performance of the protocol (such as the randomness generation rate, the probability of aborting, etc), one typically has an implementation (which may utilise imperfect and noisy devices) in mind – this is commonly referred to as the “honest implementation” of the protocol. In our case, we want to implement Protocol 1 using an (imperfect) homodyne detector and thus, the honest implementation that we consider here is a homodyne detector with imperfect detection efficiency.

9. In Fig. 3, is the electronic noise constant regardless of LO? Have they checked whether the photodiodes are saturated in high LO powers?

Yes, the electronic noise remains stable during the experiment. We also checked that the photodiodes are not saturated. In the experiment, a LO with 10 mW is used, corresponding to 5 mW input power for each photodiode. Based on the measured response of a single photodiode, as shown in Fig. 1, the photodiode is working in the linear region and does not saturate.

FIG. 1. Photodiode response. The blue circles are the measurement data, and the red line is the linear curve fitting.

10. In Fig. 4, by system efficiency, does it mean detectors efficiency or efficiency of the whole design? Including the efficiency of components used in the source and measurement sides?

The term “system efficiency” refers to the overall transmission efficiency, which takes into account all losses outside of Alice’s lab (hence, it does not account for the losses on the source side). This includes the channel losses, the photodiodes’ quantum efficiencies and the losses that are equivalent to the electronic noise of the detector.

11. **In Fig. 4, 10^{10} round experimental data is reported; how does it work, and is it averaged over all rounds?**

In Fig 4, the solid curves are results from theoretical simulation of the expected randomness expansion rate when the protocol is implemented using a homodyne detector with detection efficiency indicated in the x -axis. The theoretical model neglects all other imperfections such as bias in the beam-splitter, intensity fluctuations of the local oscillator, etc. In the experiment, we run the protocol using a homodyne detector with effective efficiency of about 91.7% for $n = 10^{10}$ rounds. The data point in Fig 4 indicates that the actual statistics that we obtained in the experiment is indeed very close to the theoretical model – the observed frequency of losing the game is within the range of the tolerated statistical fluctuation from the expected losing probability. Consequently, we can indeed certify a gross randomness generation rate (before accounting for the randomness of the inputs) of 0.00455 bits per round and an expected randomness expansion rate (after accounting for the randomness invested to choose the inputs) of 0.00199 bits per round.

12. **Minor point in Fig. 4, there are discontinuities in the figure, which could be due to numerical precision, while it could be due to some minor issues in the optimization problem. Unfortunately, based on the security analysis represented in the paper, I couldn’t reproduce the results given in Fig. 4, as the work should be reproducible; I was wondering if the authors have some guidance on the optimization problem or some repository with the codes?**

The discontinuities in Fig 4(b) is mainly the resolution issue with the plot as we only evaluate the net randomness expansion rate on a small number of detection efficiencies. This is because it took significantly longer time to evaluate the data for the 16-QAM protocol as compared to the 4-QAM protocol (the SDP scales pretty badly with the size of the matrix). As can be seen in Fig 4(a), the curves are significantly smoother as we have evaluated more data points for that figure. We do not have a repository with the codes but we can provide the codes upon request. At the same time, we also welcome any constructive criticisms on how we can describe the security analysis better so that the plots can be verified independently of our codes.

To maintain the anonymity of the reviewers, we can also provide our codes to the reviewers via the editor if the editor agrees to this. We will also provide a file (README.md) that contains the dependencies for our code and where the reviewers can download them.

13. **Very minor point is the figures’ positioning, which can be easily fixed.**

We thank the reviewer for pointing this out. We have adjusted the ordering and positioning of the figures.

14. **Finally, even though the source is trusted in this protocol, still the rate is not significantly higher than fully DI QRNG; it would be great if the author could justify this point.**

We thank the reviewer for this comment. There are multiple explanations on why our protocol have limited rate improvement as compared to the fully DI QRNG. Firstly, recall that DI and semi-DI frameworks rely on statistics to reveal the “quantumness” (to use the same term used by Reviewer 1) of the underlying implementation. In most DI QRNG protocols, they simply take the measurement outputs as it is (except possibly mapping the “no-detection” events to some deterministic output). In our protocol, however, we perform some coarse graining of the outputs: the homodyne detection gives us an almost continuous variable output but we simply map it to two possible outputs by taking the sign of the outputs. Coarse graining is, in general, bad in the DI and semi DI framework: different raw statistics can be mapped into the same coarse grained statistics and the framework has to account for the worst case explanation for the coarse grained statistics. Hence, admittedly, we do not harness the full potential of the protocol. There are practical reasons for this decision. Firstly, the size of the SDP scales badly with the number of outputs that we consider. Secondly, it is unclear how we can formulate a good P&M game with more outputs. Note that this is a limitation of our proof technique and not the limitation of the protocol itself (one could possibly improve the randomness generation rate by considering the fine-grained statistics).

Another factor that is related to the security analysis is that in formulating the min-tradeoff function, we lower bound the conditional von Neumann entropy using guessing probability. This bound is known to be suboptimal. On the other hand, there are tight analytical bounds on von Neumann entropy for DI QRNG protocols that use

the CHSH game. This is also a limitation of the security proof rather than the protocol itself. In fact, by applying the bound introduced by Brown *et al* (*arXiv*: 2106.13692) into our SDP relaxation, we can get an arbitrarily tight bound to the conditional von Neumann entropy. However, the resulting SDP is also significantly bigger than that of our current formulation and hence, it takes significantly longer time to compute the rate. As the tightness of the randomness certification is not the focus of this work (indeed, we managed to demonstrate randomness expansion even with a suboptimal bound), we think that the rate improvement via Brown *et al*'s bound can be left for future work.

Another possible factor for the lower randomness generation rate is the fact that our protocol only uses one measurement device, whereas the typical DI-QRNG protocols use two measurement devices. As such, for every round, our protocol only produces 1 weakly random bit whereas most DI-QRNG protocols produce 2 weakly random bits per round.

That being said, as pointed out by the Reviewer 1, while the throughput of our protocol is only slightly better than that of DI-QRNG protocols, the experiment is considerably more practical and there are many room for improvements (such as miniaturisation using PIC implementation as well as its ability to operate at room temperature). Significant improvement can also be obtained by optimising the constellation of test states (as shown in Fig 4(b) which has about 1 order of magnitude improvement in the net expansion rate). As such, the additional assumption on the state preparation can be justified from the standpoint of practicality.

Row	Work #1	rate	Work #2	rate	Work #3	rate	Work #4	rate
1	Ref 10	100 kbits/s	Ref 11	1 Mbits/s				
2	Ref 12	6.5 Mbits/s	Ref 13	12 Mbits/s	Ref 14	2 Gbits/s	Ref 15	3.55Gbits/s
3	Ref 16	2.9 Gbits/s	Ref 17	8 Gbits/s				
4	Ref 18	1.7 Gbits/s	Ref 19	8.2 kbits/s				
5	Ref 20	5 kbits/s						
6	Ref 21	17.42 Gbits/s	Ref 22	8.05 Gbits/s				
7	Ref 23	1.25 Mbits/s						
8	Ref 24	145.5 MHz						
9	Ref 25	theoretical work						
10	Ref 26	16.5 Mbits/s	Ref 27	230 kbits/s				
11	Ref 28	113 Mbits/s						
12	Ref 29	23 bits/s						
13	Ref 30	50 kbits/s (theoretical simulation, assuming 100 MHz repetition rate)	Ref 31	5.7 kbits/s	Ref 32	0.00012 bits/emitted photon ($\mu = 0.66$, repetition rate of optical pulses = 30 Hz)		
14	Ref 4	42 new random bits, $n = 3016$ successive entanglement events over the period of about one month	Ref 35	127.86 bits certified using the security definition of Ref 4, and 72.70 bits certified using modern security definition (based on Ref 4's experimental data)				
15	Ref 7	3606 bits/s (expansion rate)						
16	Ref 5	240 bits/s (generation rate)	Ref 6	114 bits/s (generation rate)	Ref 8	2.57E8 net bits (randomness expansion) over 19.2 hours (equiv. 3718 bits/s expansion rate with locality loophole open) and 2.63E8 net bits (randomness expansion) over 220 hours (equiv. 332 bits/s with spacelike separation)		
17	This work	4.98 kbits/s (expansion rate)						

REVIEWER COMMENTS

Reviewer #1 (Remarks to the Author):

I thank the author for considering and answering my comments. I find the answers on point. The current version of the manuscript, to me, is now eligible to be submitted as it is.

Reviewer #2 (Remarks to the Author):

Thanks to the authors for addressing my points and sending a comprehensive response letter. I still believe a general comparison of this protocol is missing—as this might be even hard for people in the QRNG community to understand the differences between semi-DI protocols, and in particular, yours, which is unique in security. Having said that, it is necessary to add a paragraph explaining the advantages of this protocol over the similar semi-DI and particularly compared to protocol with a partially trusted source, as the rate of this protocol is extremely lower while the plus could be the level of security, etc.; thus it would be vital to clarify these points for the sake of the readers which might not be expert in the field and also showing the advantages and disadvantageous of the work for possible commercial application. Based on the authors' answer, I could suggest they add a figure which compares security as a function of computational/experimental complexity versus a fully DI case. Of course, this is just a suggestion, and it's up to them how to clarify this.

It would be appreciated if the author could clarify this.

RESPONSE TO REVIEWERS' COMMENTS

Firstly, we would like to thank both reviewers for taking their time to carefully read our manuscript and give constructive feedback to our work. We have carefully considered their suggestions and made the appropriate modifications to our manuscript based on their valuable feedback. For the reviewers' convenience, the changes from the previous version will be written in blue colour. We also note that we have added Ref. [4] and Ref. [21] which may change the numbering of the references.

RESPONSE TO REVIEWER 1

I thank the author for considering and answering my comments. I find the answers on point. The current version of the manuscript, to me, is now eligible to be submitted as it is.

We thank the reviewer for his/her suggestions in this peer-review process. We are glad that the reviewer found that our manuscript is eligible to be submitted.

RESPONSE TO REVIEWER 2

Thanks to the authors for addressing my points and sending a comprehensive response letter. I still believe a general comparison of this protocol is missing—as this might be even hard for people in the QRNG community to understand the differences between semi-DI protocols, and in particular, yours, which is unique in security. Having said that, it is necessary to add a paragraph explaining the advantages of this protocol over the similar semi-DI and particularly compared to protocol with a partially trusted source, as the rate of this protocol is extremely lower while the plus could be the level of security, etc.; thus it would be vital to clarify these points for the sake of the readers which might not be expert in the field and also showing the advantages and disadvantageous of the work for possible commercial application. Based on the authors' answer, I could suggest they add a figure which compares security as a function of computational/experimental complexity versus a fully DI case. Of course, this is just a suggestion, and it's up to them how to clarify this. It would be appreciated if the author could clarify this.

We thank the reviewer for taking his/her time to carefully examine our reply in the previous correspondence. We understand the reviewer's concern that a general comparison of our protocol to the existing works is missing.

While there appears to be a natural hierarchy in comparing trusted device, semi-device-independent (DI), and fully DI frameworks based on the strength of the assumptions, it is actually not easy to establish a fair comparison among different semi-DI frameworks. This is because each semi-DI framework is based on a unique sets of assumptions, which may not be directly comparable due to their qualitative nature. For instance, it would be awkward to represent various semi-DI and DI frameworks on a plot comparing their security as a function of the experimental complexity—since there is no common consensus on how to quantitatively measure experimental complexity across implementations with different qubit systems; this is more or less the same problem when people tried to compare different quantum computing architectures. We feel, the most appropriate (and high-level) way to understand the differences between different semi-DI protocols is to elucidate their level of assumptions at the device level (i.e., trusted/untrusted source vs trusted/untrusted measurement), which is what we did in Table 1.

We also agree it would be very useful to explain in greater detail the merits of our semi-DI approach, especially how it enables the concept of quantum randomness expansion/generation with minimal assumptions in practice. To that end, we added a few more paragraphs in the discussion section, covering our method's potential to be miniaturised and how to overcome its current low-rate limitation using more measurement outcomes (instead of binary outcome) and a larger encoding scheme, i.e., 16-QAM.